# Dependence of cell fate potential and cadherin switching on the coordinate within the primitive streak during differentiation of human pluripotent stem cells

Ye Zhu[1] and Aryeh Warmflash[1,2,*,‡]

## ABSTRACT

During gastrulation, mesendoderm originates in the primitive streak (PS) where cells undergo an epithelial-mesenchymal transition and an expression switch from E- to N-cadherin. We made measurements of these processes during differentiation of human pluripotent stem cells to PS and downstream mesendoderm subtypes using established protocols and variants in which signaling through key pathways, including activin, BMP and Wnt, were modulated. The anterior-to-posterior identity of cells within the PS had little impact on the subsequent differentiation potential but impacted the degree of cadherin switching. During switching, E-cadherin downregulation and N-cadherin upregulation were uncorrelated and had different dependence on signaling. The exception to the broad potential of cells was the loss of definitive endoderm potential in mid-to-posterior PS. Thus, cells induced to different PS coordinates had similar potential within the mesoderm but differed in cadherin switching. Consistent with this, E-cadherin knockout or overexpression did not alter outcomes during differentiation. Overall, although all processes are regulated by the same set of signaling pathways, the extent of cadherin switching and epithelial-mesenchymal transition can vary substantially within cells adopting the same cell fate.

KEY WORDS: Gastrulation, Primitive streak, Signaling, Cadherin switching, EMT, Pluripotent stem cells

## INTRODUCTION

During mammalian gastrulation, the primitive streak (PS) forms and subsequently differentiates into endoderm and mesoderm subtypes. This process involves an epithelial-mesenchymal transition (EMT), during which PS cells change from epithelial to mesenchymal morphology and begin to migrate. EMT is marked by cadherin switching, whereby E-cadherin (E-CAD; also known as cadherin 1 or CDH1) is downregulated, and N-cadherin (N-CAD; also called cadherin 2 or CDH2) is upregulated (reviewed by Amack, 2021). In addition to cell adhesion, cadherins have been also shown to play a crucial role in fate commitment during neural differentiation. E-CAD downregulation actively promotes neural differentiation (Malaguti et al., 2013), while N-CAD expression helps stabilize neural fate in the neural epithelium (Punovuori et al., 2019). Nonetheless, the role of cadherin switching in mesendodermal fate commitment remains unclear.

Protocols to induce human pluripotent stem cells (hPSCs) to PS and mesendodermal subtypes, such as definitive endoderm (DE), paraxial mesoderm (PM) and lateral mesoderm (LM), have been well established using combinations of activin/Nodal/TGFβ (hereafter 'activin'), BMP, Wnt and FGF/ERK signaling activation and inhibition (Burridge et al., 2014; Gertow et al., 2013; Cheung et al., 2012; Umeda et al., 2012; Patsch et al., 2015; Bernardo et al., 2011; Martyn et al., 2019). Differentiation to any of these subtypes can be achieved with 2 days of differentiation from hPSCs, with the first day corresponding to PS induction and the second corresponding to further specialization to PM, LM or DE. Protocols to induce PS include FGF and Wnt activation along with either activin (anterior PS), BMP (posterior PS) or both (mid PS). Protocols for DE induction on day 1 are similar to those for anterior primitive streak (APS) induction with increased activin and decreased Wnt stimulation (Loh et al., 2014). Cells are thought to be partially committed to mesodermal subtypes following PS differentiation (Hsu et al., 2018; Loh et al., 2014, 2016; Nakanishi et al., 2009). DE differentiation on day 2 involves continued activin stimulation with BMP inhibition while the combination of Wnt activation and BMP inhibition induces PM from APS. Conversely, BMP activation combined with Wnt inhibition induces LM from mid primitive streak (MPS) (Loh et al., 2016) (Fig. 1A).

Although these protocols have been extensively used, the degree to which the potential of different PS populations differs as well as the relationship between cell fate, on the one hand, and EMT and cadherin switching on the other, remain unclear. To address this gap, we investigated the potency of different PS populations and cadherin switching dynamics in differentiating hPSCs. We found that different regions of the PS maintain similar potency within the mesoderm, indicating that spatial coordinates within the PS do not rigidly dictate mesodermal fate commitment. While APS cells retained potency for both endoderm and mesoderm, mid-to-posterior PS lost differentiation potential toward DE while retaining broad potency within the mesoderm. The degree of cadherin switching did correlate with PS position and was regulated by signaling independently of fate commitment. Furthermore, we observed that E-CAD downregulation and N-CAD upregulation were not strongly correlated and occurred independently during mesendoderm differentiation. Finally, loss of epithelial and gain of mesenchymal properties and cadherin switching were also more independent from each other than previously appreciated so that cells could acquire mesenchymal properties even without cadherin switching. Consistent with this, knocking out or overexpressing E-CAD resulted in only minor effects on mesendodermal differentiation. Taken together, our results suggest

[1]Department of Bioengineering, Rice University, Houston, TX 77005, USA. [2]Department of Biosciences, Rice University, Houston, TX 77005, USA.
*Present address: Division of Developmental Biology, Cincinnati Children's Hospital Medical Center, Cincinnati, OH 45229, USA.

‡Author for correspondence (aryeh.warmflash@cchmc.org)

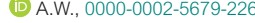 A.W., 0000-0002-5679-2268

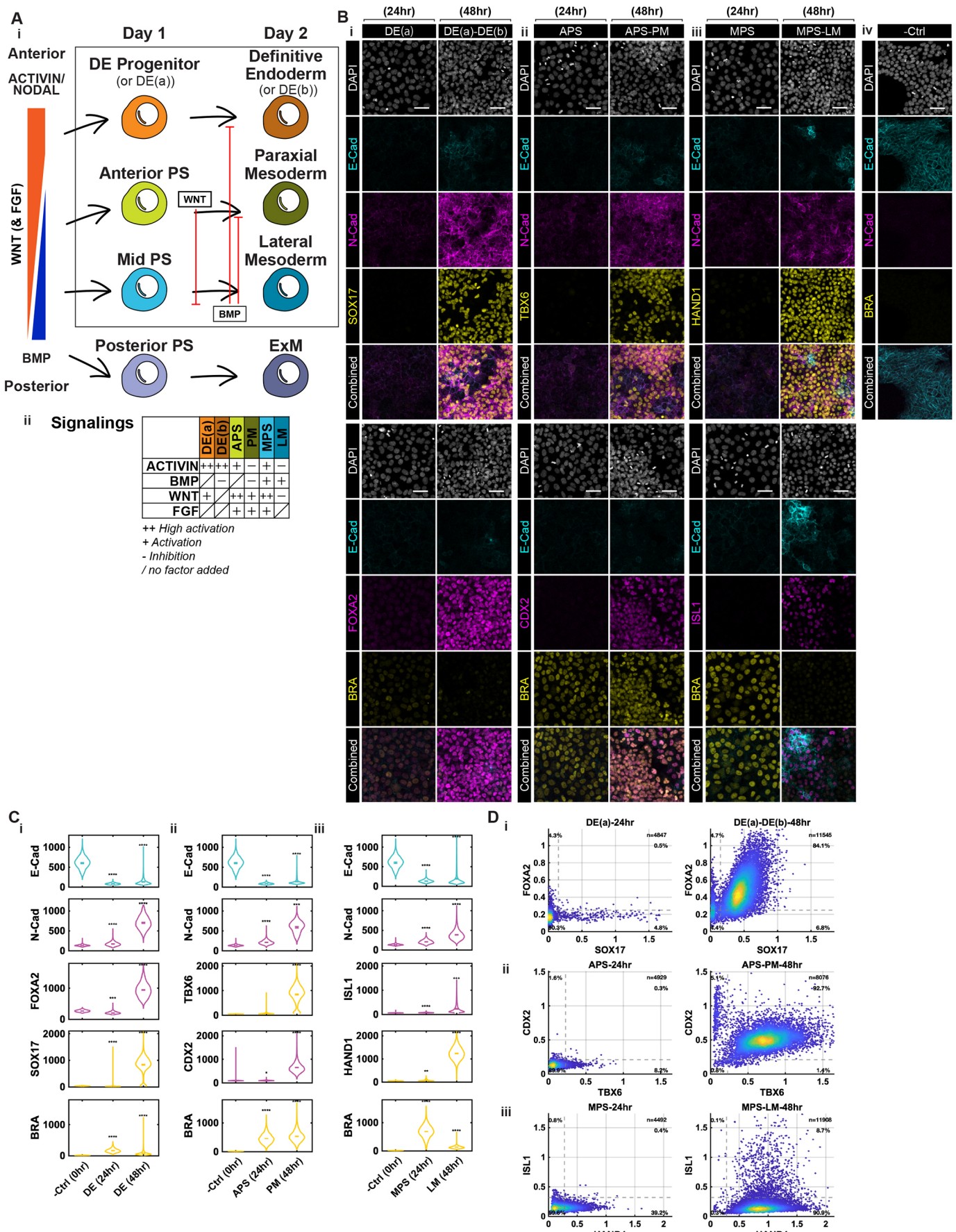

**Fig. 1.** See next page for legend.

**Fig. 1. Fate marker and cadherin expression during established protocols for DE, PM and LM.** (A) Schematic of established protocols for mesendoderm induction from hPSCs through the activin, BMP, Wnt and FGF signaling pathways. (B) Example confocal immunofluorescent images of hPSCs following the indicated treatments. Fate-specific markers are used: BRA for PS; SOX17 and FOXA2 for DE(a) and (b); TBX6 and CDX2 for APS and PM; HAND1 and ISL1 for MPS and LM; and the cadherin markers E-CAD and N-CAD. Scale bars: 50 μm. (C) Quantification of fluorescence intensity of the indicated markers following the indicated protocols. In the violin plots, black bars indicate per image means. Statistical comparisons were performed on per-image medians from two independent experiments using Welch's two-sample *t*-test with Bonferroni correction. Asterisks denote adjusted significance (*$P<0.05$, **$P<0.01$, ***$P<0.001$, ****$P<0.0001$). (D) Co-expression scatter plots of normalized mean fluorescence intensity in each cell (relative to DAPI) showing FOXA2 versus SOX17 for DE(a) and (b); CDX2 versus TBX6 for APS and PM; ISL1 versus HAND1 for MPS and LM. Each dot represents one cell (*n*=total number of cells per condition); color indicates density of overlapping points. Dashed lines denote manually defined thresholds used for quadrant separation. Percentages indicate the proportion of cells in each quadrant. Two independent experiments performed; six images per condition.

that cells throughout the PS have broad overlapping potential and that cadherin switching and EMT can vary substantially within cells adopting the same fate.

## RESULTS

### Generation of mesendodermal cell types with established protocols

Using previously reported protocols (Fig. 1A), we differentiated hPSCs into PS subtypes and mesendodermal lineages, generating DE, PM and LM within 48 h (Fig. 1B,C). In each case, we obtained a largely homogeneous population with >80% of the cells expressing expected markers: SOX17[+] and FOXA2[+] for DE, TBX6[+] and CDX2[+] for PM, and HAND1[+] for LM. In the case of LM, we observed heterogeneity in ISL1 expression with 20% of cells expressing this marker (Fig. 1D).

To understand the heterogeneity within the LM population, we stained for additional markers associated with extra-embryonic (GATA3) and cardiac (NKX2-5) fates (Fig. S1A-D). We found that nearly all ISL1[high] cells are GATA3[+], compared to only a minority of HAND1[+] cells expressing GATA3. Notably, the ISL1[high] cells are also E-CAD positive. Thus, the presence of a small extra-embryonic population (HAND1[+]/IS1[high]/GATA3[+]/E-CAD[+]) was identified, in addition to the main LM population (HAND1[+]/ISL1[low]/E-CAD[−]). While some data suggest that GATA3 is expressed in both extra-embryonic mesoderm and amnion, we found that these cells also express AP2A, which is more exclusive to the amnion (Chen et al., 2024 preprint; Chhabra and Warmflash, 2021) (Fig. S2). By contrast, NKX2-5 expression did not correlate with high ISL1 or HAND1 expression in LM: only small percentages of LM cells expressed NKX2-5; however, the LM cells were competent to generate cardiac mesoderm as >50% of cells expressed NKX2-5 when a cardiac differentiation was applied on the third day following LM induction (Fig. S1C,D).

In summary, the DE and PM populations obtained using established protocols were relatively homogeneous, with high co-expression of the key fate markers, while cells subjected to the LM protocol exhibited some heterogeneity and included a subpopulation of amnion-like cells.

### DE, PM and LM protocols have different cadherin switching dynamics

We next examined the dynamics of cadherin switching and EMT marker expression during mesendoderm differentiation. E-CAD

downregulation was observed at 12 h after induction across all PS subtypes, marking the initiation of cadherin switching, while N-CAD upregulation was not detected at 12 h but was evident at 24 h (Fig. S3A). At 12 h, Snail expression remained undetectable, suggesting that E-CAD repression can be initiated while SNAIL1 (SNAI1) expression remains low (Fig. S3A). Thus, E-CAD downregulation is an early and consistent event in mesendodermal differentiation, prior to N-CAD or EMT marker activation.

The timing and level of N-CAD upregulation and EMT marker expression varied among the three lineages within the first day of induction (Fig. S3A). DE had the lowest Snail expression, consistent with observations in mouse embryos *in vivo* that Snail1 (Snai1) expression is limited to the PS and the nascent mesoderm but not DE (Carver et al., 2001; Smith et al., 1992) and that Snail1 is not required for endoderm formation (Scheibner et al., 2021).

By 48 h, >85% of cells expressed N-CAD in all mesendodermal lineages (Fig. S4Ai,Aiii). Concurrently, a small subset of ~10-20% of cells began re-expressing E-CAD in each lineage with the highest percentage in LM (Fig. S4Ai,Aiv). Nearly all E-CAD[+] cells also expressed N-CAD. Notably, E-CAD[+] amnion cells found in the LM protocol also had high N-CAD expression (Fig. S4Ai). *In vivo*, the amnion does not express N-CAD, but our results here indicate that these cells are capable of expressing both cadherins, consistent with the more general decoupling between fate and cadherin switching we observe (see below). In short, all populations could be separated by cadherin expression into a majority of cells which were E-CAD[−]/N-CAD[+], and a small subgroup of E-CAD[+]/N-CAD[+] cells.

We utilized the markers ZO-1 (TJP1) and vimentin (VIM) to analyze whether cells adopted an epithelial or mesenchymal phenotype (Fig. S4C,D). ZO-1 is localized to tight junctions on the apical side of epithelial cells, including pluripotent cells, while VIM is an intermediate filament protein expressed in mesenchymal cells (Yang et al., 2020). At 24 h of differentiation, PS cells lost epithelial characteristics with ~10% of cells expressing ZO-1 compared to 90% in the pluripotent state. VIM expression remained low, however, with levels comparable to the pluripotent controls (Fig. S4B). By 48 h, VIM was elevated in all mesendodermal lineages with cells co-expressing N-CAD. Snail dynamics differed between the lineages with downregulation in LM and continued expression in PM (Fig. S3A,C). N-CAD[+] cells lost expression of E-CAD at the junctions and appeared mesenchymal in nature.

These observations suggest that while DE, PM and LM share a similar initiation of EMT and cadherin switching, the subsequent dynamics differ among the lineages, particularly concerning Snail expression dynamics and the fraction of cells that re-express E-CAD.

### Cells at different coordinates in the PS have similar potency in mesoderm

We then investigated whether cell differentiation to different positions within the PS, as represented by the treatments that induce APS, MPS and posterior primitive streak (PPS), exhibit distinct potencies toward PM and LM. As the PS protocols differ in the inclusion of activin and BMP, we also included a control group without addition of either ligand, labeled as the −TGFβ group (Fig. 2A).

All four groups activated BRA (brachyury; TBXT) and MIXL1 by 24 h (Fig. 2Bi). Upon further induction toward PM, all groups expressed TBX6 and CDX2, and when directed toward LM HAND1 expression was robust, indicating that APS, MPS, PPS and −TGFβ groups all retained potency for PM and LM fates

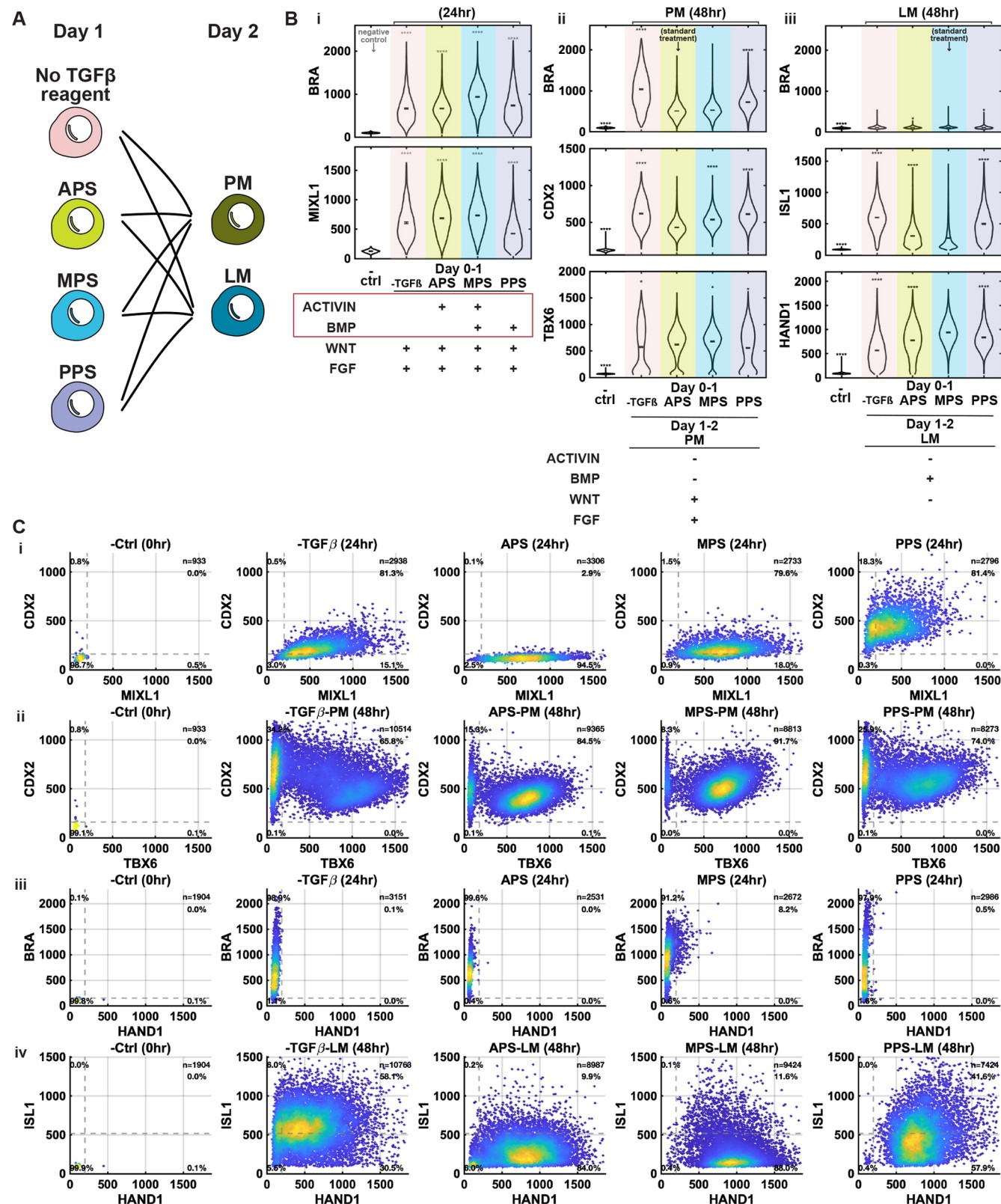

**Fig. 2.** See next page for legend.

(Fig. 2Bii, iii), although quantitatively there were minor differences in the distribution of marker expression depending on the PS induction protocol (Fig. 2C). Both the −TGFβ and PPS groups were deprived of activin A during PS induction, and these groups

displayed lower TBX6 and higher CDX2 expression in PM induction at 48 h, as well as higher ISL1 expression in LM induction. These findings suggest that activin signaling during the first 24 h helps prevent cells from committing to extra-embryonic

**Fig. 2. Cells at different coordinates in the PS have similar mesodermal potency.** (A) Schematic of the experimental design testing PM and LM potency of APS, MPS, PPS, and the treatment with no activin A or BMP4 (denoted the −TGFβ group). (B) Quantification of markers of primitive streak (BRA and MIXL1), PM (BRA, CDX2 and TBX6) and LM (BRA, ISL1 and HAND1) from confocal microscopic images of cells with the indicated treatments and fixed at the time indicated above the plots. In the violin plots, black bars indicate per-image means. Statistical comparisons were performed on per-image medians from two independent experiments using Welch's two-sample *t*-test with Bonferroni correction. Asterisks denote adjusted significance (*$P<0.05$, ****$P<0.0001$), relative to the standard treatment group. (C) Co-expression scatter plots of fate markers for the indicated treatments and fixation times. Each dot represents one cell (*n*=total number of cells per condition); color indicates density of overlapping points. Dashed lines denote manually defined thresholds used for quadrant separation. Percentages indicate the proportion of cells in each quadrant. Two independent experiments performed; eight images per condition.

fates while enhancing potential for mesodermal fates (Fig. 2B), consistent with data from modulated activin signaling during micropatterned differentiation (Jo et al., 2022). Taken together, our data suggest that, once cells adopt PS fates, they have similar potential for different fates within the mesoderm regardless of the coordinate; however, signaling that varies with position in the PS may influence the decision between extra-embryonic and PS fates with extra-embryonic fates more suppressed at anterior coordinates.

### Endoderm progenitors in PS exhibit differential potency towards mesodermal and endodermal fates

We assessed the endodermal potency of different PS subtypes and included a treatment that is similar to APS induction but has been optimized for endoderm differentiation (Loh et al., 2014) [marked as DE(a); the second day is marked as DE(b)]. Cells treated with DE(a) activated BRA, the PS/mesodermal progenitor marker, albeit at the lowest level compared to the other PS subtypes (Fig. 3B). At this stage, expression of the endodermal marker SOX17 remained undetectable. Upon extending the treatment of different PS subtypes with DE(b) on the second day, the DE(a)-DE(b) and APS-DE(b) groups showed similar co-expression of the endodermal markers SOX17 and FOXA2 (Fig. S5ii). In contrast, MPS and PPS largely failed to express SOX17 and FOXA2 (Fig. 3B). These findings suggest that the DE(a) and APS subtypes retain potency for both DE and PM, whereas the mid-to-posterior PS subtypes lose their potency for DE. To ascertain whether this represented a true loss of potency or a delay in adopting endodermal fates, we extended these protocols for another day to perform MPS/PPS-DE(b)-DE(b). The fraction of SOX17⁺FOXA2⁺ double-positive cells remained <5% (Fig. S6), suggesting that mid-to-posterior PS cells do indeed lose potency for endoderm.

We tested the mesodermal potential of cells treated with the endoderm optimized DE(a) protocol. By extending the treatment with PM induction on day 2, DE(a)-PM exhibited an identical TBX6/CDX2 expression distribution as the original APS-PM treatment. Similarly, DE(a)-LM also activated HAND1 expression, confirming the broad mesodermal potency of DE(a) cells (Fig. 3C).

In summary, cells from different coordinates in the PS (including endodermal progenitors) presented similar potency toward mesodermal subtypes. However, cells at the mid-to-posterior end of the PS lost potency toward endoderm (Fig. 3D).

### The coordinate within the PS strongly and separately affects E-CAD downregulation and N-CAD upregulation

While mesendodermal treatment groups exhibited similar potency and fate commitment across lineages, the dynamics of cadherin switching and EMT progression varied significantly across the PS subtypes from anterior to posterior. To explore these differences, we examined expression of Snail, E-CAD and N-CAD in the APS, MPS, PPS and −TGFβ groups directed towards PM and LM fates. In some experiments, to aid in quantification of E-CAD and N-CAD fluorescence, we used a cell line with a cerulean-CAAX membrane marker and improved the signal from this marker in fixed cells using an anti-GFP antibody.

During the initial 24 h as cells exited pluripotency, activin downregulated while BMP upregulated E-CAD as determined by comparing APS and PPS to the MPS and −TGFb conditions, respectively. Thus, among the PS treatments, the lowest levels of E-CAD were found in APS and the highest levels in PPS. In contrast, Snail was highest in the condition with both activin and BMP treatment, suggesting that both signals upregulate this factor (Fig. 4A, Fig. S7). At 48 h, these trends largely persisted with E-CAD highest in the posterior conditions but almost completely lost in APS and MPS conditions, and N-CAD and Snail highest in the mid-to-posterior regions (Fig. S7). Both N-CAD and Snail expression were higher in PM than LM across all PS coordinates, which may reflect a switch in the role of BMP from activating to repressing these factors between day 1 and day 2. This will be explored in more detail below.

In the absence of all added TGFβ ligands, both N-CAD and Snail activation were impaired. In the −TGFβ-LM condition, these factors were completely absent while E-CAD was strongly expressed, as BMP upregulated E-CAD expression but not Snail or N-CAD expression on day 2 (Fig. 4A, Fig. S7). In PM, Snail and N-CAD were upregulated while E-CAD was downregulated (Fig. S7), likely in response to the strong Wnt signal activation on day 2 in this condition (see below).

We also compared DE(a) and APS treatments followed by either DE(b) or PM induction on day 2. Cells treated with the APS protocol yielded lower E-CAD and higher Snail and N-CAD compared to DE(a) at the end of day 1 with this trend still present but less pronounced on day 2 (Fig. 4B-D). This likely reflects the higher concentration of the Wnt agonist CHIR99021 in APS compared to DE(a) media as Wnt signaling promotes both cadherin switching and Snail expression (see below).

Taken together, these data suggest that signals which vary with the PS coordinate affect cadherin switching and Snail expression on both the first and second day. On day 1, during PS induction, activin downregulates, while BMP maintains, E-CAD expression, while both of these signals upregulate N-CAD and Snail expression. Thus, while N-CAD and Snail expression closely mirror each other, the expression of E-CAD and N-CAD are largely decoupled due to their differential regulation by activin signaling. By day 2, the expression of all three of these markers can vary within cells of the same cell fate, as their expression, but not the cell fate, depends on the day 1 conditions (Figs 2C, 4B-D, Fig. S7).

While these data suggest that the levels of cadherin can vary substantially within cells adopting the same fate, it remains true that the same signaling pathways control cadherin switching and cell fate decisions, which may induce some constraints on the expression levels within each fate. To address this issue directly, we performed the protocols above and triple stained for E-CAD, N-CAD and fate markers and examined the fate markers as a function of E-CAD and N-CAD expression (Figs S9, S10).

We found that cells induced to LM displayed a wide variety of E-CAD and N-CAD levels and these largely did not correlate with expression of markers of amnion or LM, including GATA3, AP2A, HAND1 and CDX2 (Fig. S10). The main exception was AP2A, which primarily showed high E-CAD expression levels; however, some

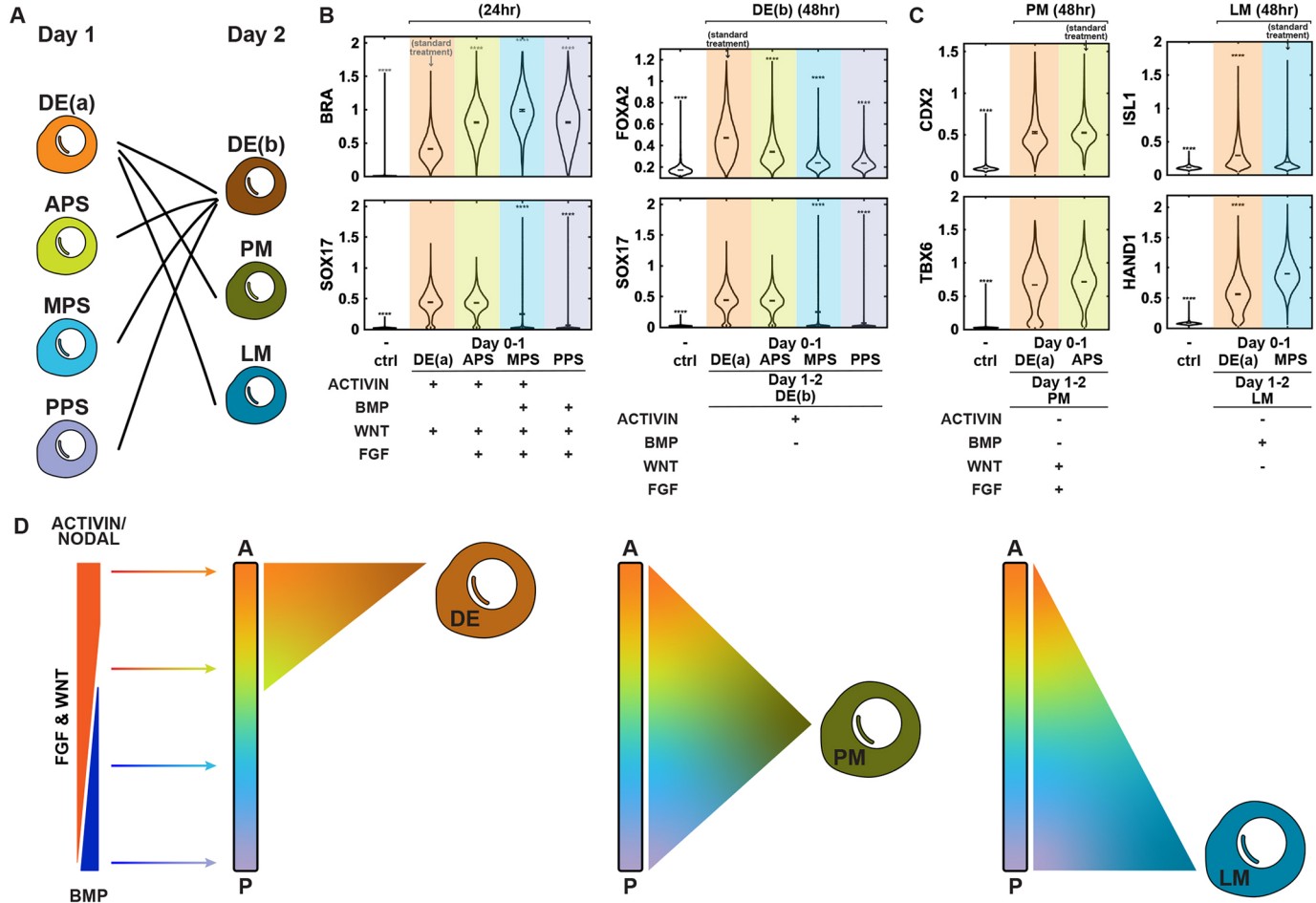

**Fig. 3. Anterior PS cells have broad potential while mid and posterior PS cells lose DE potential.** (A) Schematic of the experimental design testing the DE potency of APS, MPS, PPS, and mesodermal potency of DE(a) treated cells. (B) Quantification of endodermal markers from confocal microscopic images taken following the indicated treatments and fixation times. (C) Quantification of PM and LM comparing standard protocols and those in which an endodermal differentiation protocol is followed on day 1. In the violin plots in B,C, black bars indicate per-image means. Statistical comparisons were performed on per-image medians from two independent experiments using Welch's two-sample *t*-test with Bonferroni correction. Asterisks denote adjusted significance (*P<0.05, **P<0.01, ***P<0.001, ****P<0.0001), relative to the standard treatment group. Two independent experiments performed; six images per condition. (D) Schematic indicating the range of PS cell types with the potential to give rise to each further differentiated cell type. A, anterior; P, posterior.

AP2A$^+$ cells also had lower E-CAD or high levels of N-CAD. These data support that both amnion and LM can range from E-CAD$^+$N-CAD$^-$ to E-CAD$^-$NCAD$^-$ and many possibilities in between. Cells differentiated to PM fates also showed widely varying E-CAD and N-CAD levels (Fig. S9). Here, the majority of cells expressed CDX2 regardless of cadherin levels, while TBX6 expression was more restricted to a particular region of the E-CAD/N-CAD plane. Essentially, all TBX6 cells were N-CAD$^+$; however, some TBX6 cells had low levels of N-CAD, and TBX6$^+$ cells ranged from not expressing E-CAD to expressing substantial levels of E-CAD.

In summary, all cell fates were consistent with substantial variations in cadherin levels. LM cells were consistent with any level of cadherin expression, TBX6$^+$ PM cells all expressed some N-CAD but could express E-CAD, and all AP2A$^+$ amnion cells expressed E-CAD but some also expressed N-CAD. For the required expression (N-CAD in PM, E-CAD in amnion), expression levels were highly variable.

## Transcriptome analysis confirms broad potential of PS progenitors

To validate our results on PS potential at the transcriptome level, we performed RNA sequencing of samples differentiated to the four PS populations [DE(a), APS, MPS and PPS] at 24 h as well as each of these populations further differentiated to LM, PM or DE(b). Hierarchical clustering of all samples grouped samples based on the day 2 treatment without regard to the PS induction protocol on day 1, indicating that generally all the PS progenitors are capable of yielding populations with similar transcriptomes when further differentiated (Fig. 5A). Consistent with this, performing principal component analysis (PCA) and grouping samples on the PC1 versus PC2 plot by day 2 treatment showed coherent groups of conditions, while grouping based on day 1 treatment did not (Fig. 5B).

At the level of individual genes, we confirmed that fate markers such as *TBX6* were expressed at similar levels regardless of the initial PS differentiation (Fig. 5C). *SOX17* was also expressed at similar levels in all PS protocols, followed by DE(b) except for PPS-DE(b), while *FOXA2* was low in both MPS-DE(b) and PPS-DE(b), consistent with our results above that MPS and PPS lose the potential for DE differentiation (Fig. S11). The cadherin genes *CDH1* and *CDH2* showed variability within each fate with, in general, high levels of *CDH1* (E-CAD) expression when the initial treatment was either DE(a) or PPS compared with the APS or MPS treatments. As in the immunostaining data, *CDH2* (N-CAD)

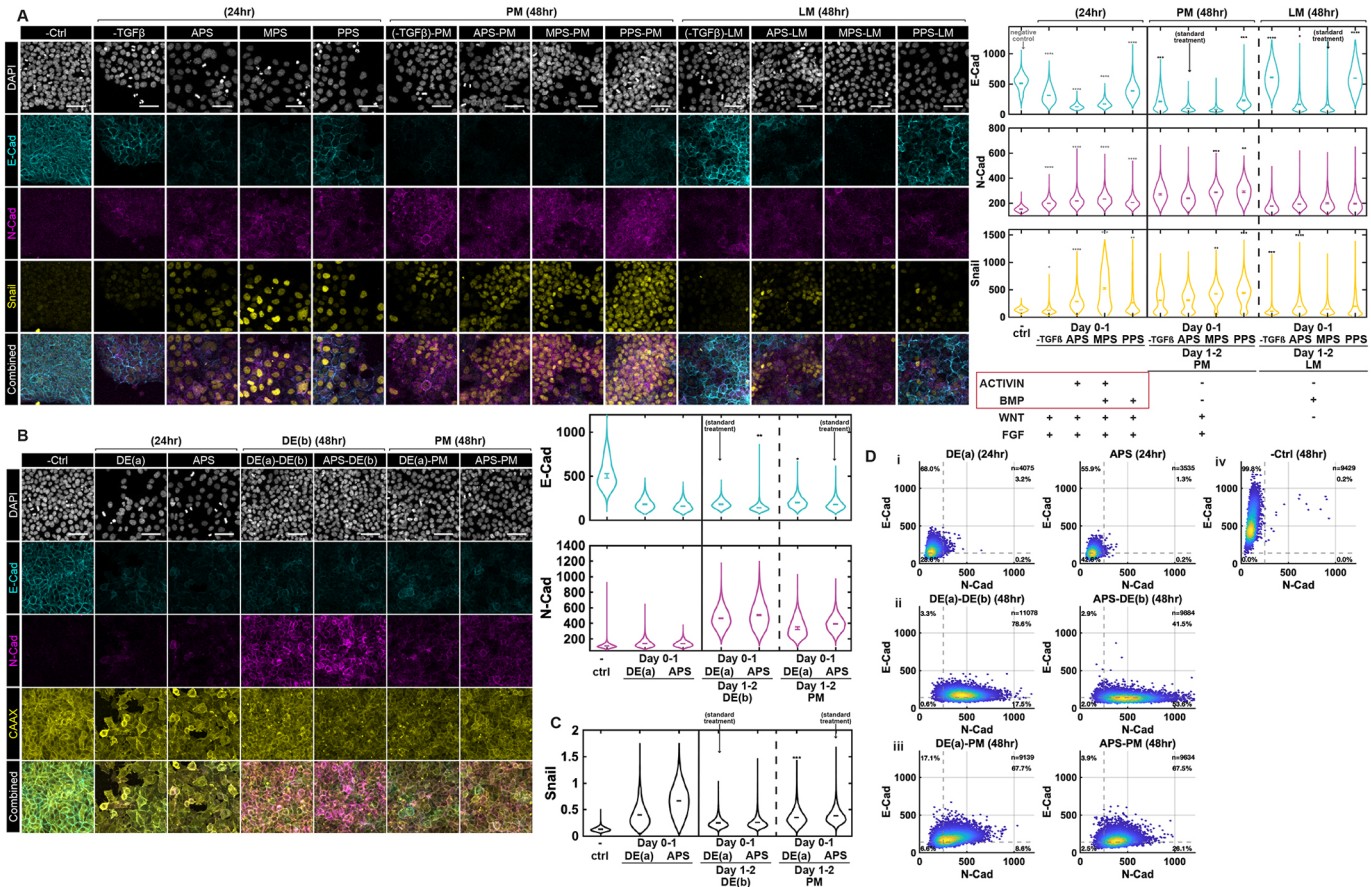

**Fig. 4. Cadherin expression varies considerably within cells adopting the same fate.** (A) Example confocal immunofluorescence images and quantification based on fluorescence intensity of E-CAD, N-CAD and Snail for the indicated treatments and fixation times. (B,C). Example confocal immunofluorescence images and quantification based on fluorescence intensity of E-CAD and N-CAD (B) or Snail (C) comparing standard protocols for PM with those in which DE(a) was followed on day 2. ESI-017-CAAX:mCerulean line and anti-GFP were used to visualize cell membranes (Fig. S8). In the violin plots, black bars indicate per-image means. Statistical comparisons were performed on per-image medians from two independent experiments using Welch's two-sample *t*-test with Bonferroni correction. Asterisks denote adjusted significance (**P*<0.05, ***P*<0.01, ****P*<0.001, *****P*<0.0001), relative to the standard treatment group. (D) Co-expression scatter plots of E-CAD and N-CAD comparing using DE(a) or APS on day 1 of differentiation followed by either PM or DE(b). Each dot represents one cell (*n*=total number of cells per condition); color indicates density of overlapping points. Dashed lines denote manually defined thresholds used for quadrant separation. Percentages indicate the proportion of cells in each quadrant. Two independent experiments performed; six images per condition. Scale bars: 50 µm.

showed different trends depending on cell fates with levels increasing towards the posterior PS in LM differentiation but decreasing in DE or PM treatment (Fig. 5C). Notably, PPS-DE(b) and MPS-DE(b) clustered closer to DE(a)-DE(b) and APS-DE(b) than to other fates, suggesting global similarity between the transcriptome of cells with these treatments. However, as noted above, both conditions failed to express *FOXA2*, and PPS-DE(b) cells failed to express *SOX17* as well. This is consistent with our immunostaining results above that showed very few FOXA2⁺SOX17⁺ double-positive cells under these conditions. Thus, we conclude that mid-to-posterior PS cells lose endoderm potential, but when treated with DE(b) they do maintain transcriptomic similarity with DE cells. More rigorously determining the potential of all these conditions would require going beyond gene expression to examine the potential of cells differentiated to endodermal subfates, which is an interesting topic for future investigation.

## Activin promotes, and BMP inhibits, cadherin switching during MPS-LM induction

To better visualize and quantify cadherin dynamics in time during mesendodermal differentiation, we generated a dual-reporter hPSC line, with fluorescent tags at the endogenous loci for E-CAD and N-CAD and a DOX-inducible membrane fluorescent marker for cell identification (ESI-017 E-CAD: mCitrine N-CAD:mCherry CAAX:mCerulean; Fig. 6A, Movie 1). This reporter cell line maintained pluripotency and exhibited mesendodermal induction potency comparable to the wild-type (WT) parental line (Fig. S12B-D). We focused on modulating the MPS-LM protocol as MPS induction media contains all four signals involved in PS induction.

Live-cell imaging during MPS-LM induction captured the progression of cadherin switching. E-CAD downregulation initiated early, while the reporter allowed us to capture gradual upregulation of N-CAD that began almost immediately upon differentiation and continued throughout the first day. N-CAD upregulation accelerated on the second day and peaked at around 30 h, and then downregulated and stabilized by 48 h (Fig. 6B). These rapid changes likely reflect the switch from MPS to LM media and highlight the dynamic nature of cadherin expression in response to signaling during mesodermal differentiation.

To elucidate the roles of activin and BMP signaling in this process, the MPS-LM induction protocol was modified by withdrawing activin

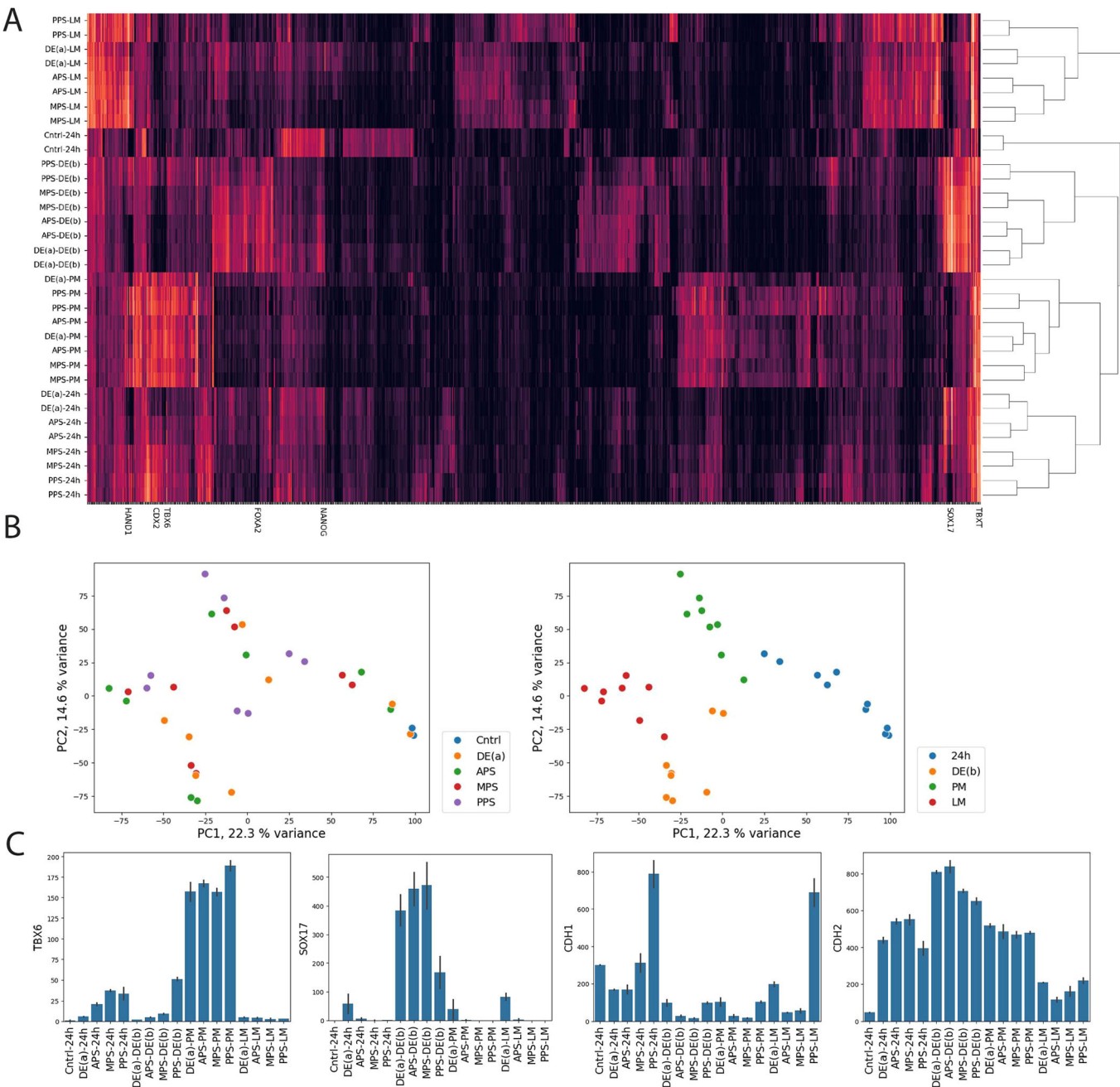

**Fig. 5. Transcriptomic analysis confirms broad potential of PS cells.** (A) Hierarchical clustering of RNA sequencing data. Some of the marker genes used in this study are indicated. (B) Principal component plot of RNA sequencing data colored by either the PS induction protocol performed on day 1 (left) or the differentiation protocol on day 2 (right). (C) Expression of individual genes in RNA-sequencing samples.

A or BMP4 (Fig. 6B). Notably, activin had a stronger effect on cadherin dynamics than BMP signaling. Removing activin A impaired both E-CAD downregulation and N-CAD upregulation, demonstrating its crucial role in promoting cadherin switching. In contrast, withdrawing BMP4 had minimal impact on E-CAD downregulation but caused N-CAD to stabilize at a higher level by 48 h, confirming the inhibitory role of BMP on N-CAD upregulation on the second day of differentiation, consistent with our results above. In summary, activin and BMP have opposing roles in regulating cadherin switching during lateral mesodermal induction, whereby activin promotes cadherin switching and BMP selectively inhibits N-CAD expression on day 2 (Fig. 6D).

## Roles of activin, BMP and Wnt doses during differentiation

We further investigated the effects of activin signaling in initiation of EMT and cadherin switching, as well as fate specification for LM induction by either adding the activin inhibitor A-83-01 or varying the dose of activin A (0, 10, 30 or 100 ng/ml; with 30 ng/ml activin A being the original dosage).

On day 1, withdrawing activin A or adding the inhibitor A-83-01 impaired E-CAD downregulation and Snail and N-CAD upregulation (Fig. 6C), emphasizing the crucial role of exogenous activin signaling in initiating EMT and downregulating E-CAD during early PS induction. By continuing these groups with the original second-day LM induction protocol, we observed that, when

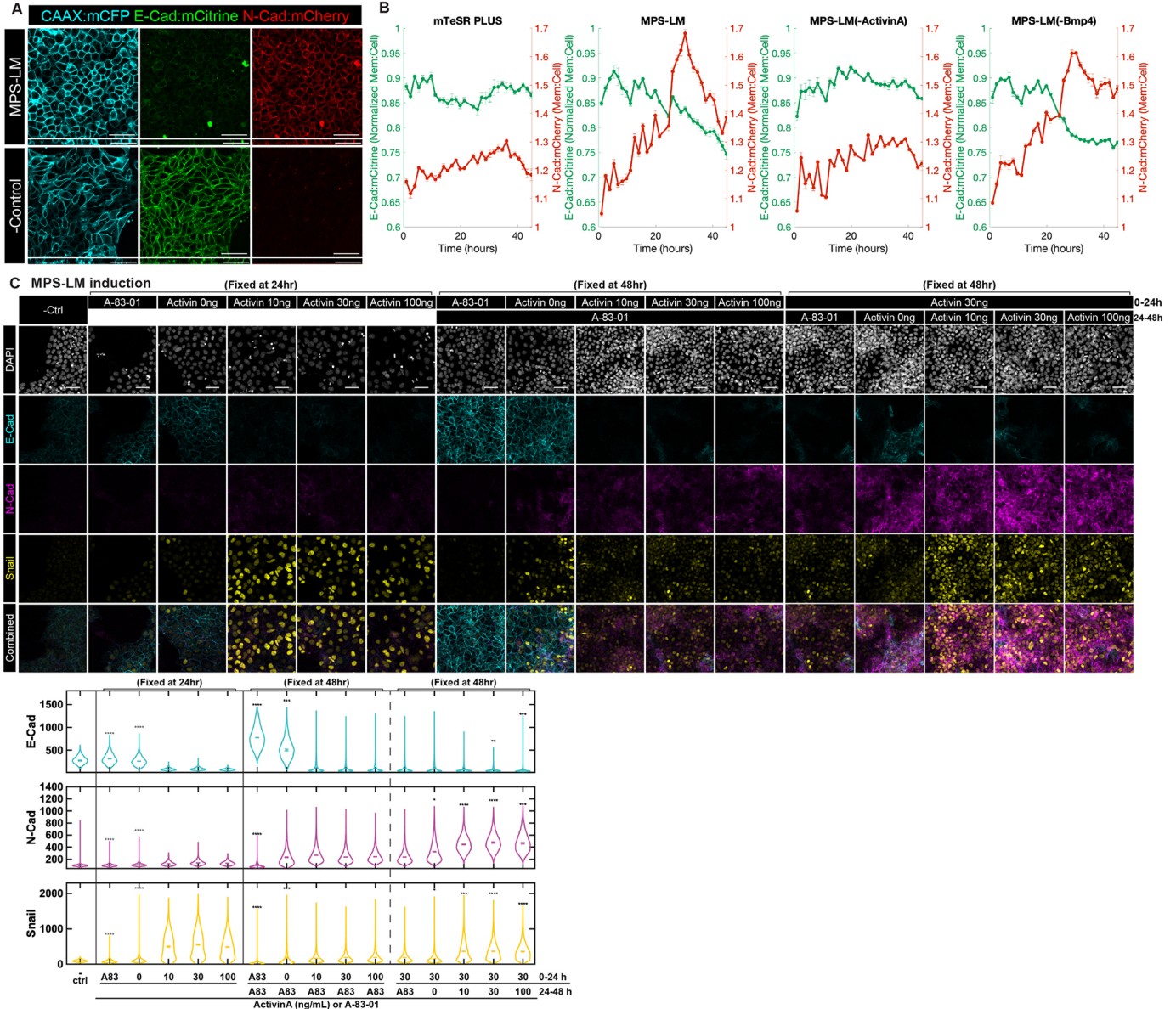

**Fig. 6. Cadherin switching is regulated by activin and BMP signaling.** (A) Example confocal microscopy images of live ESI017-CDH1:mCitrine-CDH2:mCherry-CAAX:mCerulean reporter cell line after 48 h MPS-LM treatment or mTeSR Plus control treatment. (B) Quantification of the fluorescence intensity of CDH1:mCitrine and CDH2:mCherry in cells with the indicated treatments (0-44 h). N=3 positions per condition. (C) Example confocal immunofluorescence images and quantification based on fluorescence intensity of E-CAD, N-CAD and Snail for cells that underwent MPS or MPS-LM induction with either A-83-01 (1 µM) or various activin A doses (0, 10, 30 and 100 ng/ml) and fixed by 24 h or 48 h, as indicated. In the violin plots, black bars indicate per-image means. Images from the standard differentiation condition at 48 hours (activin 30 ng/ml followed by A-83-01) are shown within each series (columns 10 and 12) for ease of comparison. Statistical comparisons were performed on per-image medians from two independent experiments using Welch's two-sample t-test with Bonferroni correction. Asterisks denote adjusted significance (*P<0.05, **P<0.01, ***P<0.001, ****P<0.0001), relative to the standard treatment group. Two independent experiments performed; seven images per condition. Scale bars: 50 µm.

activin was included, varying the dosage did not significantly alter LM fate commitment (HAND1+/ISL1low) or cadherin dynamics (Fig. 6C, Fig. S13). Inhibition of activin/Nodal signaling caused E-CAD to remain highly expressed and a failure to express N-CAD, while simply removing the exogenous activin caused co-expression of E-CAD and N-CAD across the population. The effects on cell fate were less pronounced; however, the absence of activin signaling on the first day promoted GATA3 and ISL1 expression, increasing the fraction of cells in the extra-embryonic lineage (Fig. S13A). Nonetheless, there was still substantial mesoderm differentiation as reflected in BRA expression, with the resulting population a mixture

of amnion and mesoderm (Fig. S13B). These results show that activin signaling is required for both E-CAD downregulation and N-CAD upregulation while it quantitatively biases cells to mesodermal rather than amnion lineages, but is not strictly required.

Removing A-83-01 from the day 2 LM protocol significantly enhanced N-CAD expression without changing the expression of fate markers. Adding exogenous activin further enhanced N-CAD expression, and diverted cells away from the LM lineage as reflected in a dose-dependent loss of HAND1 expression (Fig. 6C, Fig. S13). In summary, activin signaling on the first day is essential for initiating EMT, downregulating E-CAD and

suppressing extra-embryonic fates, while its role on the second day is primarily to promote N-CAD upregulation. High levels of activin signaling during day 2 are not consistent with LM fate commitment. Similar experiments varying the activin concentration on the second day in the APS-PM protocol showed that activin signaling suppressed CDX2 and TBX6 expression, consistent with a diversion away from the PM fate, but had little effect on N-CAD expression (Fig. S14).

We also investigated modulating BMP signaling levels during the MPS-LM protocol and found only minor effects of BMP dosage on cell fate or cadherin switching. Most significantly, inhibiting BMP on day 1 resulted in lower expression of both N-CAD and E-CAD expression while somewhat reducing potency of cells for LM, as reflected in a fraction of cells failing to express HAND1. In contrast, inhibiting BMP on day 2 compromised LM induction but had only minor impacts on cadherin switching (Fig. S15A,B).

We similarly investigated the role of Wnt signaling in the APS-PM protocol. Removing CHIR99021 or adding the Wnt inhibitor IWP2 on day 1 compromised E-CAD downregulation, Snail and N-CAD upregulation, and cell fate specification, while inhibiting it on day 2 impaired PM differentiation with smaller effects on cadherin or Snail expression (Fig. S16A,B). Thus, Wnt signaling is essential for all aspects of PS formation, EMT and cadherin switching on day 1, and for PM differentiation on day 2.

### E-cadherin knockout or overexpression has minor impacts on fate induction and lineage-specific cadherin dynamics

Our results above suggest that cell fate acquisition and EMT are largely independent of cadherin switching while N-CAD upregulation and E-CAD downregulation are also separate processes. To validate these results functionally, we created an E-CAD knockout (KO) cell line through CRISPR-Cas9 RNP delivery. The loss of E-CAD was validated by sequencing and immunostaining (Fig. 7A, Fig. S17A). Notably, E-CAD KO cells exhibited localized ZO-1 expression to the apical side, showing an epithelial morphology with apical-basal polarity (Fig. 7A). Expression of the pluripotency markers SOX2, OCT4 (POU5F1) and NANOG in the E-CAD KO cells remained comparable to that of WT cells (Fig. S17C).

Differentiation of E-CAD KO cells was similar to WT in PS induction (MIXL1, BRA) conditions, and the subsequent DE (SOX17), PM (TBX6) and LM (HAND1) differentiation (Fig. 7B). E-CAD KO cells differentiated to different PS subtypes maintained lineage potency toward both PM and LM fates just as WT cells, with indistinguishable TBX6 and HAND1 expression between the cell lines (Fig. 7C, Fig. S18). These results suggest that removal of E-CAD does not affect the cell fate specification processes directly.

We also examined N-CAD and Snail expression in E-CAD KO cells and noted minor differences compared with WT (Fig. S19). Snail expression was slightly but consistently lower in E-CAD KO compared to WT cells, while variable differences were noted in N-CAD expression. In particular, E-CAD KO cells showed reduced N-CAD during DE differentiation but increased N-CAD during LM differentiation. Taken together, these results suggest that E-CAD is not essential for PS and subsequently endodermal and mesodermal differentiation. It is also not required for Snail or N-CAD expression, but its loss changes the quantitative expression of these factors in a manner that depends upon the lineage.

To understand the effects of loss of E-CAD more globally, we performed RNA sequencing of E-CAD KO cells in pluripotent cells as well as in the standard DE(a)-DE(b), APS-PM and MPS-LM protocols. Examining global gene expression in hierarchically clustered samples showed that each KO sample clustered with the

corresponding treated WT sample and had highly similar gene expression (Fig. S20A). Examining the samples in a PCA plot, we found that similarly treated samples aligned along PC1 regardless of genotype; however, the E-CAD KO samples were consistently shifted along PC2 (Fig. S20B). We examined a panel of developmental genes and found nearly identical patterns of expression in ECAD KO and WT cells (Fig. S20C). To better understand the differences induced by ECAD KO, we examined the top 50 upregulated and downregulated genes in each comparison (e.g. control treated WT versus E-CAD KO cells). The genes identified were mostly not known to have roles in PS development, and so we used gget (Luebbert and Pachter, 2023) to query gene ontology (GO) databases with this list of genes. The GO terms were largely not directly related to developmental processes, for example the top GO terms for genes upregulated by the E-CAD knockout in DE differentiation were 'mRNA processing' and 'mRNA splicing' while those downregulated by E-CAD KO were associated with 'pyruvate metabolic processes' and 'glycolytic processes'. One interesting exception was the downregulation of several genes associated with Wnt signaling upon loss of ECAD in APS-PM differentiation, including *LGR5*, *NOTUM*, *DKK1* and *DKK4*, which includes both positive and negative regulators of Wnt signaling. Nonetheless, the overall effect was not sufficient to substantially change cell fate markers. Thus, these data show that while loss of E-CAD does have effects on the cell biology of differentiating hPSCs, genes associated with differentiation and cell fate are not affected. We provide lists of differentially expressed genes and associated GO terms comparing ECAD KO with WT cells in all conditions in Tables S3 and S4.

It is possible that the lack of developmental effects of the loss of E-CAD is due to compensation by upregulation of a different cadherin. We thus examined expression of cadherins and protocadherins as well as ephrin genes, which are also involved in cell adhesion (Fig. S21). The results confirmed strong downregulation of *CDH1* at the RNA level in the KO, but showed few other differences between WT and E-CAD KO. *PCDH8* was upregulated in the E-CAD KO specifically in the APS-PM protocol, while *PCDHB5* was expressed at lower levels in all E-CAD KO samples compared to WT. The significance of these changes in gene expression is an interesting topic for future study; however, these results establish that there is likely minimal compensation by other members of the cadherin family upon loss of E-CAD in this context.

Finally, we examined the effects of overexpressing E-CAD on differentiating hPSCs. We lipofected hPSCs with an E-CAD-GFP fusion plasmid without selecting for transfected cells, allowing us to compare transfected and un-transfected cells in the same field of view. We found that transfected GFP$^+$ cells expressed markers of differentiation including TBX6 in the APS-PM protocol, SOX17 in the DE(a)-DE(b) protocol, and HAND1 in the MPS-LM protocol at levels that were not distinguishable from their GFP$^-$ neighbors (Fig. S22A-C). The expression of N-CAD was also minimally affected by E-CAD overexpression (Fig. S22D). Thus, overexpressing E-CAD did not cause substantial changes in expression of cell fate markers or N-cadherin.

### DISCUSSION

In this study, we used hPSCs to examine the dynamics of differentiation, cadherin switching and EMT during mesendoderm differentiation. The first day of the protocol involves exit from pluripotency and upregulation of PS markers and is generally considered to specify the coordinate within the PS. The second day's treatment typically dictates the subset of mesendoderm such

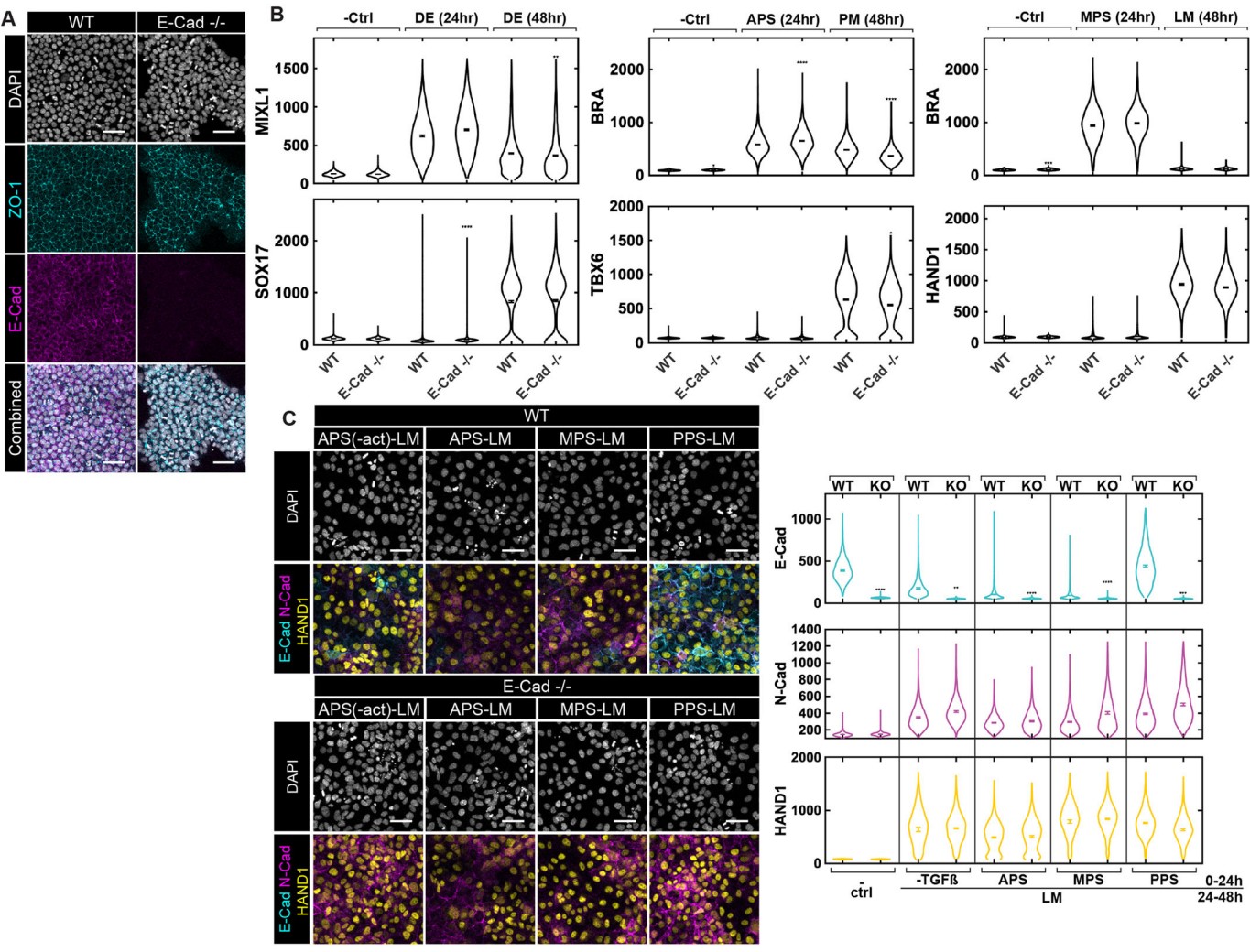

**Fig. 7. *CDH1* KO has minor effects on fate commitment in mesendoderm differentiation.** (A) Example confocal microscopic images of ESI017 wild type (WT) and E-CAD KO (E-cad $^{-/-}$) (maintained in mTeSR Plus) immunostained for DAPI, ZO-1 and E-CAD. Scale bars: 50 μm. (B). Quantification based on fluorescence intensity of ESI017 WT and E-CAD KO following the indicated standard protocols. '– Ctrl' indicates negative control that is maintained in mTeSR Plus and fixed at 0 h. Two independent experiments performed; eight images per condition. (C) Example confocal microscopic images and quantification of E-CAD, N-CAD, and fate markers for WT and E-CAD KO cells with varying PS induction treatment on day 1 and induced to LM on day 2. Scale bars: 50 μm. '– Ctrl' indicates negative control that is maintained in mTeSR Plus. Two independent experiments performed; five images per condition. In the violin plots, black bars indicate per-image means. Statistical comparisons were performed on per-image medians from two independent experiments using Welch's two-sample *t*-test with Bonferroni correction. Asterisks denote adjusted significance (*$P<0.01$; **$P<0.001$; ***$P<0.0001$; ****$P<0.0001$), relative to the standard treatment group.

as endoderm, LM or PM. We found that cells with different PS coordinates had broad and overlapping potential so that, with a few exceptions, the cell fate was determined by the treatment on the second day. In contrast, cadherin expression depended on signaling conditions on both days of treatment and was more graded with cells adopting each cell fate capable of expressing varying levels of both E-CAD and N-CAD. Moreover, the degree of E-CAD downregulation and N-CAD upregulation varied independently, and cadherin expression was not strictly linked to expression of EMT regulators, such as Snail, or to markers of epithelial or mesenchymal properties, such as ZO-1 and VIM. Taken together, our results suggest the events that occur in a coordinated fashion during gastrulation *in vivo* are regulated separately and can be modulated independently *in vitro*.

*In vivo* fate mapping has shown that mesendoderm subtypes emerge from particular PS coordinates, such as LM from the posterior PS and endoderm from the anterior PS. Based on these

known lineages, *in vitro* protocols were developed to regionalize cells within the PS on the first day before subsequent differentiation to particular mesendodermal fates on later days (Loh et al., 2016); however, this study did not compare different PS coordinates for their potency to generate downstream subtypes. Another study compared mesoderm induced with activin and BMP to that induced by Wnt and concluded that these have different potential (Mendjan et al., 2014), but neither of these conditions correspond directly to different PS coordinates: in the activin+BMP case, because Wnt was not added, and in the Wnt only case because no TGFβ signaling was included. Moreover, this study concluded that Wnt signaling could be inhibited in the activin+BMP context without compromising mesoderm induction, whereas our results here (Fig. S16) as well as those from a number of *in vivo* and *in vitro* studies (Chhabra et al., 2019; Huelsken et al., 2000; Liu et al., 1999; Martyn et al., 2018) argue that Wnt is indispensable for mesendoderm differentiation. Thus, it is difficult to draw

conclusions from this study regarding the potential of cells from different PS coordinates.

Our findings here suggest that cells throughout the PS maintain broad but not identical potential. All PS subtypes we tested were capable of efficiently giving rise to both LM and PM. By contrast, mid-to-posterior PS cells did not give rise to endoderm. Although not one of our PS conditions, in our studies of BMP dosage we also found that cells with inhibition of BMP type I receptor kinases by LDN-193189 on day 1 were impaired in giving rise to LM. It remains to be determined whether this situation exists *in vivo* or is an artifact of these culture conditions. Altogether, our studies suggest that cells throughout the PS have broad, overlapping, but not universal potential for mesendodermal fates. Our results are more consistent with the idea that anterior-posterior coordinate and mesendoderm potency exist on a spectrum rather than a rigid delineation between anterior and posterior progenitors that is the assumption of earlier work (Loh et al., 2016; Mendjan et al., 2014).

Our conclusions about cell fate potential are based on gene expression measurements, both of the entire transcriptome with RNA sequencing as well as evaluation of known key cell fate markers. We determined that cells have potential for a particular fate only when they showed overall transcriptomic similarity to the standard protocols for that fate and when they also expressed key markers. For example, in the case of posterior mesoderm or LM induced to DE, although there is transcriptomic similarity with established DE protocols they fail to express key endodermal markers. In the future, it will be important to truly establish the potential of cells by performing further differentiation to endodermal subtypes.

During gastrulation, mesoderm differentiation in the PS is accompanied by a switch from E- to N-CAD expression and acquisition of mesenchymal migratory properties. In *Drosophila*, embryos lacking either E-CAD (Shg), N-CAD (Cadherin-N) or both undergo normal mesoderm differentiation, patterning and morphogenesis, indicating cadherins are not required for either process in this context; however, overexpression of E-CAD can lead to defects due to inhibition of Wnt signaling through sequestration of β-catenin (Schäfer et al., 2014). In mice, evidence from KO phenotypes suggests that cadherin migration and differentiation may be separable. For example, mesoderm differentiation occurs in *Eomes* KO mice but cells fail to undergo EMT, although Snail1 expression is not affected (Arnold et al., 2008). Similarly, mutants for FGFR1 can specify mesoderm but show an accumulation of cells in the PS, indicating a failure of cells to undergo EMT. It was suggested that a pathway in which FGF signaling upregulates Snail1, which in turn downregulates E-cadherin is important for EMT in this context (Ciruna and Rossant, 2001; Deng et al., 1994; Yamaguchi et al., 1994). In all of these cases, however, mesoderm patterning was also affected with particular subsets of mesoderm enhanced at the expense of others, which may have resulted from dysregulation of signaling. For example, mouse *Fgfr1* mutants have attenuated Wnt signaling, which can be relieved through blocking of E-CAD (Ciruna and Rossant, 2001). *In vitro* studies provide an opportunity to decouple the effects of cadherins from changes in the signaling environment.

It is important to note that other cadherins are present during gastrulation and may play roles in parallel with E- and N-cadherin. Our RNA-sequencing data show expression of many members of the cadherin and proto-cadherin gene families. For example, both PCDH1 (protocadherin 1) and CDH3 (P-cadherin) expression decline during differentiation (Fig. S11), and it would be interesting to study whether any of these play roles in gastrulation stage EMT. Nonetheless, sequencing of E-CAD KO cells shows little difference

in expression of other cadherins, suggesting that these do not compensate for E-CAD upon loss.

Our results from *in vitro* experiments suggest that cadherin switching can be modulated substantially within a given cell fate. However, as the same signaling pathways are involved in regulating cadherins and cell fates, some constraints on the range of cadherin expression remain. For example, cells differentiated to different PS populations (APS, MPS or PPS) and then subjected to a PM differentiation protocol on day 2 all display a large fraction of TBX6$^+$CDX2$^+$ double-positive PM progenitors (Fig. 3). However, compared with the published APS-PM protocol, the MPS-PM protocol yields stronger N-CAD expression, while the PPS-PM protocol yields high levels of both E-CAD and N-CAD (Fig. 4). While E-CAD expression varied in these protocols from absent to substantial expression, N-CAD was always present in PM differentiation, likely reflecting the role of ongoing Wnt signaling both in promoting PM differentiation and N-CAD expression. Nonetheless, there is sufficient flexibility in the degree of signaling required for PM fates to modulate N-CAD levels several fold while retaining the same cell fate.

These results also suggest that cadherin switching should not be considered a single coordinated process but rather two separate events, E-CAD downregulation and N-CAD upregulation, which need not occur together. Treatments such as PPS-PM yield high levels of E-CAD$^+$N-CAD$^+$ cells, and in many conditions, the expression levels of E-CAD and N-CAD are correlated rather than anticorrelated at the single-cell level (Fig. S7). Even for established protocols that underwent cadherin switching, the degree of switching was highly variable (Fig. S3C). As these protocols were optimized to generate cell fates, but not to mimic *in vivo* levels of cadherin switching, and, to our knowledge, quantitative measurements of cadherin expression in space and time have not been performed *in vivo*, it remains to be determined whether these differences are reflected in the embryo.

We modulated signaling and measured Snail and cadherin expression to determine the relationship between them. On the first day, all pathways examined appeared to promote Snail and N-CAD expression, perhaps reflecting the activity of all these key pathways in initiating mesendodermal differentiation. The highest levels of Snail and N-CAD were found in MPS conditions, which include activation of all four signals. In contrast, E-CAD retention was promoted by BMP, so that E-CAD expression was highest in PPS and lowest in APS. Interestingly, E-CAD downregulation was detected earlier than Snail upregulation, indicating that other factors downstream of key pathways such as activin signaling are likely involved. The requirement for both activin and Wnt signaling in downregulating E-CAD is consistent with the established synergy between these factors in initiating PS differentiation (Yoney et al., 2018). In particular, as activin/Nodal signaling is involved in maintaining pluripotency (James et al., 2005; Vallier et al., 2005), combinatorial signaling is required to initiate mesendoderm differentiation (Singh et al., 2012; Yoney et al., 2018). On the second day of differentiation, the effects of modulating signaling on cadherin switching were generally smaller than on the first day. Elevating activin or Wnt expression within the MPS-LM protocol continued to promote Snail and N-CAD expression while BMP did not. LM differentiation, which is performed in the presence of BMP, Wnt inhibition and Nodal inhibition yielded the lowest levels of Snail and N-CAD. Interestingly, although activin signaling promotes cadherin N-CAD and Snail expression, recent studies in mouse have shown that endodermal cells, which have high levels of activin, may only partially undergo EMT (Scheibner et al., 2021).

Understanding this discrepancy will require further investigation, however, as the effects of signaling on N-CAD and Snail1 expression diminish in time; it may be that ongoing activin signaling after initial PS differentiation could promote endoderm differentiation without fully triggering EMT. Alternatively, the range of signals leading to endoderm may allow for widely varying cadherin expression as we have shown here for other fates.

Taken together, our *in vitro* studies suggest two key points. First, cells within the PS exhibit a wide window of potential which shifts gradually from anterior to poster. Second, EMT and cadherin switching can be decoupled from cell fate acquisition and are regulated independently by signaling pathways. Future research will be needed to determine how these processes, which can be regulated separately, are tightly coupled during gastrulation *in vivo*.

## MATERIALS AND METHODS
### Routine cell culture
All experiments were performed using the ESI017 hPSC line (XX; obtained from ESI BIO; RRID: CVCL_B854) or cell lines based on ESI017 (listed in Table S1). Cells were cultured in the chemically defined medium mTeSR1 or mTeSR Plus (STEMCELL Technologies) in Matrigel-coated (Corning; 1:200 in PBS–, overnight at 4°C) 35 mm or 60 mm culture dishes and kept at 37°C, 5% $CO_2$. Cells were routinely passaged using Dispase [STEMCELL Technologies; 1:5 dilution in DMEM/F12 (VWR)] and checked for *Mycoplasma* contamination with all negative results. In all experiments, cell passage number did not extend beyond 58. Single-cell suspensions were prepared using Accutase (Corning). The ROCK-inhibitor Y-27672 (10 μM; STEMCELL Technologies) was used to maintain single-cell status after seeding and before the start of treatment, and 5 μg/ml puromycin (Gibco) was used for selecting the CAAX:mCFP cell membrane marker.

### Differentiation
For differentiation, pluripotent hPSCs were prepared into a single-cell suspension using Accutase and seeded onto a 96-well round μ-Plate or 18-well chambered μ-Slide (ibidi) with No. 1.5 polymer coverslip bottom, allowing high-resolution imaging. Before seeding, the plates or slides were coated with Matrigel (1:200 in PBS–, overnight at 4°C) or with Laminin-521 (Biolamina; 1:20 in PBS++, 2 h at 37°C).

DE induction protocol was adapted from Loh et al. (2014). Anterior and mid PS, and paraxial and LM induction protocols were adapted from Loh et al. (2016). The adapted protocols are described below.

### First day treatment
After 12 h or overnight, cells were differentiated into: (1) DE progenitor [or DE(a)] using 100 ng/ml activin A+2 μM CHIR99021 in Essential 6 medium (or E6; Gibco); (2) APS using 30 ng/ml activin A+20 ng/ml bFGF+4 μM CHIR99021 in E6; (3) MPS using 30 ng/ml activin A+20 ng/ml bFGF+40 ng/ml Bmp4+6 μM CHIR99021 in E6; or (4) PPS using 20 ng/ml bFGF+40 ng/ml Bmp4+6 μM CHIR99021 in E6. Some cells were exposed to PS treatment without TGFβ reagents (noted as −TGFβ) using 20 ng/ml bFGF+4 μM CHIR99021 in E6, or with reagents as described in the text. After 24-h induction, cells were either further differentiated as described below, or fixed for immunostaining and imaging.

### Second day treatment
After first day treatment, (1) DE(a) cells were further induced into DE [or DE(b)] using 100 ng/ml activin A +250 nM LDN-193189; (2) APS cells were further induced into PM using 1 μM A-8301+20 ng/ml bFGF+250 nM LDN-193189; (3) MPS cells were further induced into LM using 1 μM A-8301+30 ng/ml Bmp4+4 μM IWP2.

The first day and second day treatment were mixed to test the potency of each fate. After 48 h, cells were fixed for immunostaining and imaging.

### Immunostaining
Cells were fixed with 4% paraformaldehyde (Electron Microscopy Science) in PBS for 20 min at room temperature and washed twice with PBS. Fixed samples were then permeabilized and blocked with blocking buffer [PBS with 0.1% Triton X-100 (Sigma-Aldrich)+3% donkey serum (Sigma-Aldrich)] for 1 h at room temperature or overnight at 4°C. Primary antibodies were diluted in the blocking buffer at the dilutions stated in Table S1, and applied to cells overnight at 4°C. After three washes using PBST [PBS with 0.1% Tween 20 (Sigma-Aldrich)], diluted secondary antibody (1:500 in blocking buffer; see Table S1) with DAPI (1:5000) were applied to cells for 45 min to 1 h at room temperature. After another three PBST washes, cells were maintained in PBS ready for imaging.

### Fixed cell imaging
Fixed cells were imaged on Olympus FV3000 laser scanning confocal microscope (LSM) with FV31S-SW v2.4.1.198 software and a 20×, NA 0.75 objective or a 40×, NA 1.25 silicon oil objective. Five to eight positions were imaged for each condition.

### Live cell imaging
Doxycycline hyclate solution (Sigma-Aldrich) was added to medium 1 day before imaging to induce CAAX:mCerulean. Dual-reporter cells were seeded in μ-Plates or μ-Slides as described above and imaged on Olympus FV3000 LSM with environmental chamber (temperature at 37°C, humidity at ~50%, and $CO_2$ at 5%), and a 40×, NA 1.25 silicon oil objective. Three positions were imaged for each condition.

### Image analysis
All experiments were performed at least twice with consistent results. 'n' in the figure captions denotes the number of positions imaged per condition in the same experiment. In all cases, error bars represent standard error of the mean intensity.

For fixed cell imaging, maximum intensity projections were generated for each image using Fiji (Schindelin et al., 2012). Nuclear masks were created by pixel segmentation of the DAPI channel using ilastik (Berg et al., 2019), and subsequent analysis was performed using custom code written using MATLAB. Nuclear protein expression was measured as the mean nuclear intensity for each cell. Membrane protein expression was quantified using directional profiling, whereby fluorescence intensities were sampled along six equally spaced directions from the centroid of each nucleus. For each direction, the top 20% fluorescence intensity values within 25 pixels from the nucleus were averaged, and the final signal for each cell was calculated as the mean of all directional values.

For live cell imaging using the ESI-017 E-CAD:mCitrine N-CAD:mCherry CAAX:mCerulean reporter cell line, pixel shift alignment was applied to align the three channels. A membrane mask and a non-membrane cell mask were generated based on the CAAX:mCerulean channel by pixel segmentation in ilastik. Background subtraction was then performed on the two membrane protein channels. Membrane protein expression was quantified as the ratio of mean membrane intensity to non-membrane cell intensity for each image.

MATLAB code is available at: https://github.com/warmflashlab/ZhuDevelopment2026.

### Establishing the ESI-017 E-CAD:mCitrine N-CAD:mCherry CAAX:mCerulean reporter cell line
The single reporter cell line ESI017 E-CAD:mCitrine was previously developed as described by Liu et al. (2022). All plasmid constructions, DNA nucleofection, and antibiotic selection followed methods elaborated by Liu et al. (2022) but using different fluorescence proteins and targeting different genes. PCR primer sequences used in plasmid construction and for genotyping can be found in Table S2.

### Plasmid constructions
N-CAD:mCherry

To insert mCherry into *CDH2* (N-cadherin) exon 16, a Cas9 and sgRNA co-expression vector, px459 (Addgene plasmid #62988) was modified to target *CDH2* (N-cadherin) exon 16 (c), with sgRNA spacer sequence 5′-ACTGAACTTCAGGGTGAACT-3′. A donor vector was created for homology directed DNA repair as follows: ~400 bp of CDH2 pro-domain or mature domain sequence was inserted into the FloxP-neomycin resistant

vector (AW-P216) as right/left homology arm, respectively, flanking the FloxP-neomycin expression cassette and mCherry. Cre Shine plasmid (Addgene plasmid #37404) was then used for excision of FloxP fragments after antibiotic-resistance selection.

### CAAX:mCerulean

Restriction enzymes (BamHI and BsrGI) were used to digest the plasmids ePiggyBac-based doxycycline-inducible cell membrane marker mCherry-CAAX (plasmid AW-P224; Liu et al., 2022) and pBSSK+-puroR-T2A-mCerulean (plasmid AW-P130). The backbone from AW-P224 and the mCerulean insert from AW-P130 were gel extracted and ligated using T4 DNA ligase resulting in a replacement of the mCherry sequence in AW-P224 with mCerulean. The doxycycline-inducible cell membrane marker mCerulean-CAAX plasmid (denoted AW-P235) was then transformed using NEB 10-beta competent *Escherichia coli* C3019H.

### DNA nucleofection

All DNA nucleofections were performed using Amaxa P3 Primary Cell 4D-Nucleofector X Kit L (Lonza).

### N-CAD:mCherry

Insertion of mCherry to the N-cadherin locus was performed by nucleofecting ESI017 E-CAD:mCitrine cells with the guide RNA plasmid, PX459-NCAD-sgRNA, together with the homology donor plasmid NCAD-mCherry-HDR. This resulted in the insertion of mCherry on the C terminus of N-cadherin followed by a PGK promoter and a neomycin resistance gene. The PGK promoter and resistance gene were flanked by loxP sites for excision. Cells were selected using G418 at 150 µg/ml for 1 week. Then CRE Shine plasmid was nucleofected to remove the PGK promoter and antibiotic resistance gene. A clonal cell line was isolated and the genotype was confirmed by sequencing.

### CAAX:mCerulean

mCerulean membrane-labeled cells were created by nucleofecting ESI-017, ESI017 E-CAD$^{-/-}$ or ESI-017 E-CAD:mCitrine-N-CAD:mCherry cells with the ePB-P-TT-mCerulean-CAAX plasmid together with the ePB helper plasmid (gifted by Ali Brivanlou, Rockefeller University, NY, USA). Cells were selected using 5 µg/ml puromycin.

### Establishing CDH1 (E-cadherin) KO cell line

The E-CAD KO cell line with ESI-017 was established via CRISPR-Cas9 RNP (ribonucleoprotein) delivery (Park et al., 2022). Nucleofection was conducted using the Amaxa P3 Primary Cell 4D-Nucleofector X Kit S (Lonza), supplemented with SpCas9 Nuclease (IDT). sgRNA spacer (sequence 5′-G*U*G*AAUUUUGAAGAUUGCAC −3′) was designed using Synthego CRISPR Design Tool and produced by Synthego. The modified cells were sorted by single-cell flow cytometry using a SH800S Cell Sorter (Sony) with Cell Sorter software (v2.2.4.5150) into individual wells of 96-well plates. Sorted cells were maintained in mTeSR Plus medium supplemented with 1× CloneR (STEMCELL Technologies). Single clones were established and then screened by Sanger sequencing (Fig. S10). Selected candidates were further verified by sequencing individual alleles with TOPO cloning (Invitrogen).

### E-cadherin over-expression transfection

The pcDNA3-E-cadherin-GFP construct (Addgene plasmid #28009) was used to induce E-CAD over-expression. ESI-017 cells were seeded at 14,000 cells per well onto Geltrex-coated µ-Slide 18 Well Glass Bottom chambers (ibidi) on Day −2. After 6 h of attachment, cells were transfected using Lipofectamine™ 3000 Transfection Reagent (Thermo Fisher Scientific) according to the manufacturer's protocol, with a final per-well mixture containing 0.15 µl Lipofectamine 3000, 0.2 µl P3000 reagent and 100 ng plasmid DNA diluted in Opti-MEM to 10 µl total volume. The DNA–lipid complex was added directly to the culture media without media change. The following day (Day −1), media was changed to remove transfection reagents and standard differentiation treatments were initiated on Day 0. GFP fluorescence confirmed transfection efficiency and E-cadherin over-expression in fixed cells by confocal microscopy.

### RNA extraction

hPSCs were differentiated for either 1 or 2 days as indicated. Cells were harvested using Accutase (Corning) and pelleted by centrifuging at 1000 rpm (214 *g*) for 4 min. The supernatant was thoroughly removed by aspiration, and cell pellets were immediately disrupted in Lysis/Binding solution (RNAqueous kit, Invitrogen). Then, samples were snap-frozen in liquid nitrogen and stored at −80°C until RNA extraction. Total RNA was isolated using the RNAqueous™-Micro Total RNA Isolation Kit (Invitrogen) following the manufacturer's protocol. Extracted RNA samples were transferred to fresh RNase-free tubes and stored at −20°C before sequencing. Bulk RNA sequencing was performed by Novogene Corporation Inc.

### RNA-sequencing data processing and quantification

Raw RNA-sequencing data were processed to obtain gene-level quantification of transcript abundance. Reads were aligned and quantified using Salmon (v1.10.2) (Patro et al., 2017) in quasi-mapping mode with sequence-specific and GC bias correction enabled to improve accuracy. A precomputed transcriptome index was used for quantification, based on the GCA_000001405.15 GRCh38_no_alt_analysis_set reference from NCBI (hg38 alias), generated using the selective alignment method (salmon_sa_index:default). Output files containing transcript-level abundance estimates were generated for each sample.

To integrate transcript annotations and facilitate downstream analysis, the quantification data were imported into R (v4.4.2), where transcript abundance estimates were summarized at the gene level. Metadata linking each sample to its experimental condition were incorporated to maintain experimental context. The resulting gene-level count matrix was exported as a CSV file for further analysis.

### Statistical analysis

Quantification was performed at a single-cell level. Results were summarized per image using the median as the representative value per image. Group comparisons were made on these per-image median values using Welch's two-sample *t*-test, with *P*-values adjusted by Bonferroni method. Unless otherwise specified, the standard treatment group was used as the control; for KO experiments, the WT ESI-017 cell line served as the control. Statistical significance is denoted as: *$P<0.05$, **$P<0.01$, ***$P<0.001$, ****$P<0.0001$.

### Acknowledgements

We thank Sally Lowell, Guillaume Blin, Idse Heemskerk and members of the Warmflash lab for helpful discussions. We thank Elena Camacho Aguilar for adapting the image processing scripts for this work. We also thank Siqi Du for optimizing the CRISPR-Cas9 RNP delivery method for efficient knockout in hPSCs.

### Competing interests

A.W. is a co-founder of and holds equity in Simbryo Technologies. Y.Z. declares no competing interests.

### Author contributions

Conceptualization: Y.Z., A.W.; Funding acquisition: A.W.; Investigation: Y.Z.; Supervision: A.W.; Visualization: Y.Z.; Writing – original draft: Y.Z., A.W.; Writing – review & editing: Y.Z., A.W.

### Funding

This work was supported by the grants from the National Institute of General Medical Sciences (R35GM149328 to A.W.), the National Institute of Child Health and Human Development (R01HD112488 to A.W.), the National Science Foundation (MCB-2135296) and Rice University (Edinburgh Rice award). Open Access funding provided by Rice University. Deposited in PMC for immediate release.

### Data and resource availability

Gene expression data from this study have been deposited in NCBI Gene Expression Omnibus under accession number GSE317192. MATLAB and Python code is available at: https://github.com/warmflashlab/ZhuDevelopment2026. Other relevant data and details of resources can be found within the article and its supplementary information. Large datasets consisting of original images are available from the corresponding author by request.

**Peer review history**
The peer review history is available online at https://journals.biologists.com/dev/lookup/doi/10.1242/dev.204807.reviewer-comments.pdf

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
