## [Peer Review File · Development (Cambridge, England)]

Dependence of cell fate potential and cadherin switching on the coordinate within the primitive streak during differentiation of human pluripotent stem cells

Ye Zhu and Aryeh Warmflash

DOI: 10.1242/dev.204807

Editor: Peter Rugg-Gunn

Review timeline

Original submission:	25 March 2025
Editorial decision:	6 May 2025
First revision received:	30 October 2025
Editorial decision:	24 November 2025
Second revision received:	17 February 2026
Editorial decision:	19 February 2026
Third revision received:	24 February 2026
Accepted:	25 February 2026

Original submission

First decision letter

MS ID#: dev.204807

MS TITLE: Dependence of cell fate potential and cadherin switching on primitive streak coordinate during differentiation of human pluripotent stem cells

AUTHORS: Ye Zhu and Aryeh Warmflash

Dear Dr Warmflash,

I have now received all the referees' reports on the above manuscript, and have reached a decision. The referees' comments are appended below, or you can access them online: please go to:

As you will see, the referees express considerable interest in your work, but have some significant criticisms and recommend a substantial revision of your manuscript before we can consider publication. If you are able to revise the manuscript along the lines suggested, which may involve further experiments and analyses, I will be happy receive a revised version of the manuscript. Your revised paper will be re-reviewed by one or more of the original referees, and acceptance of your manuscript will depend on your addressing satisfactorily the reviewers' major concerns. Please also note that Development will normally permit only one round of major revision. If it would be helpful, you are welcome to contact us to discuss your revision in greater detail.

Please attend to all of the reviewers' comments and ensure that you clearly highlight all changes made in the revised manuscript. Please avoid using 'Tracked changes' in Word files as these are lost in PDF conversion. I should be grateful if you would also provide a point-by-point response detailing how you have dealt with the points raised by the reviewers in the 'Response to Reviewers' box. If you do not agree with any of their criticisms or suggestions please explain clearly why this is so.

Reviewer 1

Advance summary and potential significance to field

The article by Zhu et al explores the dependence of cell potential and cadherin switching on primitive streak coordinates using in vitro differentiation of human pluripotent stem cells as a model. The main conclusions from this work are that cells with different primitive streak coordinates have a broad potential to form different primitive streak fates and that EMT/cadherin switching and fate acquisition can be uncoupled. The experiments have been carefully designed and quantified, and therefore the results are supported by the data. Moreover, the conclusions are novel. I support the publication of this article but have a few comments that hopefully will help improve the clarity of the manuscript:

1. The identity of the extra-embryonic mesoderm cells that are mentioned at the beginning of the result section is not clear and the manuscript has contradictory statements. In some places the authors conclude that these cells are either amnion or extra-embryonic mesoderm, and in some other places they take for granted these are amnion cells. Moreover, they also mention these cells co-express E-Cadherin and N-Cadherin while amnion cells do not express N-Cadherin and extra-embryonic mesoderm cells do not express E-Cadherin. Finally, they state that both amnion cells and extra-embryonic mesoderm cells express GATA3 but this is probably not the case. Based on Pham et al, 2022 (in vitro) and Tyser et al, 2021 (in vivo) extra-embryonic mesoderm cells do not express GATA3. This is also the case in the human reference atlas built by the Lanner and Petropoulos groups (Zhao et al, 2024).
2. The uncoupling of cell identity and Cadherin switching is very interesting. This is clearly demonstrated in Figure 5B. Nonetheless, the authors could perform a differential gene expression analysis to determine whether any other gene expression modules or signaling pathways are affected when cadherin expression is different within a given cellular identity.
3. Figure S1A shows E-Cadherin downregulation before over Snail1 expression. What may be triggering the initial E-Cadherin downregulation in this context? This could at least be discussed.
4. The authors present some data that support the idea that E-Cadherin downregulation and N-Cadherin upregulation may be independent. However, this conclusion could be strengthened with additional experiments. For example, if Activin-A is necessary for E-Cadherin downregulation during the first 24 hours, what would happen if Activin-A is added on the second day? Would this promote N-Cadherin upregulation without any changes in E-Cadherin? Along similar lines, while the E-Cadherin KO is informative, the opposite experiment (E-Cadherin upregulation) would also be interesting. Would N-Cadherin expression be affected under this condition?
5. Both Wnt and Activin-A are necessary for E-Cadherin downregulation. The authors should at least discuss whether these two signals may be working independently or within the same pathway.
6. E-Cadherin elimination typically triggers the upregulation of other cadherin types as a compensatory mechanism. For example, in the epidermis E-Cadherin depletion triggers P-Cadherin expression (Tinkle et al, 2004). As a control, the authors should check the expression of different cadherins in their E-Cadherin KO cells, as this could explain the intact apicobasal polarity that they report.

Reviewer 2*Advance summary and potential significance to field*

In this manuscript, the authors investigate the relationship between cell fates and the changes in E-Cadherin and N-Cadherin expression known to accompany differentiation in the primitive streak. The authors use a nice framework of hPSC differentiation to mimic cell fate allocation towards distinct antero-posterior identities and then use quantitative imaging to correlate cell fate markers and cadherin expression under various conditions.

By switching the signalling environment of the cells, they find that cell fate plasticity is maintained after 24h of induction except for cells induced towards mid and posterior streak identities which lose the ability to form the most anterior fate. They go on to show that E-Cadherin and N-cadherin are regulated independently from one another by signalling pathways and can be decoupled from cell fate acquisition. Finally, they show that Ecad knockout has no effect on PS cell fate.

Overall, the reasons for the dynamic regulation of Ecad and Ncad during gastrulation remain unclear. The manuscript by Ye Zhu and Aryeh Warmflash presents an interesting study that uses a nice approach to delineate the respective contributions of signalling and cell differentiation on EMT markers. I think the data will be of interest to the readership of Development but there are some areas that require improvement to meet the standards of the journal:

In my opinion, some claims are not fully supported by the data, in particular the notion that Ecad/Ncad regulation can be decoupled from cell fate. Data presentation would benefit from tightening up and the transcriptomic analyses should be further exploited to refine the message of the paper.

Comments for the author

Major:

1) Cell fate characterisation:

* "Cells subjected to the LM protocol exhibited some heterogeneity and included a small subpopulation of amnion-like cells"

The fact that this population displays heterogeneity is important for interpretation of the data shown later in the manuscript (see my comment number 2). The authors, tested the presence of amnion and cardiac progenitors (Fig S3) - which is useful - but pluripotent cells primed with WNT and FGF and then subjected to BMP may also produce other mesodermal derivatives such as PAX2+ intermediate mesodermal cells and TBX6+ PXM cells. It would be important to further characterise heterogeneity in this condition, using other stainings (TBX6, PAX2 for example). The authors could also exploit their RNAseq dataset shown later in the manuscript, focusing here on the fate markers expressed in each condition.

* "In vivo, the amnion does not express N-Cad, and the extraembryonic mesoderm does not express E-Cad, but our results here indicate cells adopting one of these fates are capable of expressing both cadherins"

This is interesting and it would be important to determine what exactly is this Ecad+/Ncad+ double positive population. Could this reflect asynchrony in the rate of differentiation in the culture? Is this population maintained at day 3? Please comment.

2) Cell fate plasticity

In Fig3, the authors show that MPS and PPS lose potency to form DE. Is this really the case or are the cells delayed in redirecting to the DE fate? This could be addressed by looking at marker expression at day 3.

3) Correlation between fate and Ecad/Ncad expression:

One of the main messages of the authors is that fate and EMT markers can be decoupled. This idea rests mostly on the comparison between Fig 2 and Fig 4 "By day 2, the expression of all three of these markers can vary within cells of the same cell fate, as their expression, but not the cell fate, depends on the day 1 conditions." (Minor: it would be helpful if the authors could reference the corresponding figures here)

I don't think this is well supported by the data: Fig 4A shows only marginal variation in Ncad levels in the fates redirected to LM. Furthermore Ncad seems to correlate closely with the levels of CDX2 in the fates redirected to PM (Fig 4A and 2B).

The main change occurs with Ecad levels which are elevated in PPS → LM, and TGFβ- → LM. These conditions correspond to the situations where the population is heterogeneous and enriched for

ISL1+ cells (Fig. 2 iv). PPS→LM (higher ISL1 + higher Ecad), -TGFb →LM (higher ISL1 and higher Ecad).

These observations contradict the authors claim. Perhaps a more in depth analysis of their co-staining of Ecad with ISL1 (Fig S2) would help clarifying this point - Is there indeed a lack of correlation between these 2 markers at the single cell level? The same should be done with CDX2 and Ecad. Complementary to this, and to circumvent the technical limitations of image quantification of membrane markers, the authors could use their double reporter line, sort out the cells based on levels of Ecad/Ncad and check the levels of fate markers by qPCR.

4) Influence of signalling on the levels of Ecad/Ncad:

*"During the initial 24 hours as cells exited pluripotency, Activin downregulated while BMP upregulated E-Cad"

I am not sure where is the evidence that BMP upregulates Ecad, the highest levels are found in the Ctrl where there is no BMP treatment.

*"both Snail and N-Cad expression were highest in the condition with both Activin and BMP treatment, suggesting that both signals contribute to upregulating these factors"

I agree for Snail but this is not so clear for NCad.

* "cells with APS yielded lower E-Cad and higher Snail and N-Cad compared to DE(a) (Fig. 4B-D)"

I agree for Snail at day 1 but I can't see any significant difference at day 2.

Please comment on the three points above. Overall, it is difficult to decouple fate from signalling since signalling dictates fate. I think the authors should discuss this and clarify which data exactly demonstrate that they were able to uncouple the relative contribution of differentiation and signalling on EMT markers. Alternatively, they could moderate the claim "that EMT and cadherin switching can be decoupled from cell fate acquisition and are regulated independently by signaling pathways" and acknowledge the limitations of the study in the discussion section. It would also be useful if the authors could discuss how to reconcile the idea that Activin promotes Ecad downregulation while other authors have shown that endoderm (an activin-driven cell fate) maintains Ecad and only undergo partial EMT (Scheibner et al)

5) Transcriptomic data:

"Transcriptome analysis confirms broad potential of PS progenitors" This claim rests on the observation that hierarchical clustering regroups treatments on day 2 regardless of day 1 conditions. I am not sure if this is a strong argument to confirm the earlier results (Fig. 2-3). In particular, all PS populations further differentiated towards DE cluster together even though the authors argued that MPS and PPS lost ability to make DE. The transcriptomic profile of PPS-LM also seem to differ significantly from the other samples.

This dataset probably contains lots of useful information and a more in depth analysis is needed. For example, it would be helpful to see the DEG between different PS conditions further differentiated towards the same fate. Such analyses will clarify differences and similarities between treatments.

6) Clarity, consistency and stats:

* I found it difficult to understand the logical progression between figures at times. For example, in Fig 1, the authors introduce their differentiation scheme and characterise marker expression for DE, APS, and MPS (PPS is shown in the diagram but not tested in the actual experiment). Then, Fig 2A (about fate plasticity) drops DE, tests PPS and introduce a new TGFb- condition. It would improve consistency and completeness if all the differentiation conditions were introduced and characterised in Fig 1.

* The manuscript includes quantifications and most data points are shown - which is great - but the legends do not state how many independent experiments were performed and there are no statistical test in any of the figures. This is needed to assess the robustness of the data and help with interpretation. The scatter plots could also include proportions as % in the quadrants of the

plots. If these represent triple staining on different plots, it would help to show the proportions of each subpopulation in a table or in a bar chart or pie chart.

7) It is interesting that the Ecad KO does not alter cell fate. The manuscript would benefit from additional discussion regarding potential compensation mechanisms upon Ecad KO.

Minor:

* Fig. 1 - the difference between the DE and APS conditions is a bit unclear, is it simple that FGF is not included in the DE condition or are the concentrations in WNT and Activin different?

* Fig. S1 and S2 are mentioned after S3 in the text.

* I found it inconvenient to switch between figures to compare fate markers and EMT markers (Fig 2 and 4 for example). The authors might consider reorganising the manuscript to show the results of the switch to PM on one figure containing both fate and EMT markers and then another figure showing the switch to LPM.

* Fig 4B has AntiGFP signal, this is not commented and not explained what GFP represents here

Reviewer 3

Advance summary and potential significance to field

Using in vitro differentiation of human pluripotent stem cells (hPSCs) toward mesendoderm fate, the authors investigate the interdependence of (1) intermediate cell state at 24 hours (corresponding to different anterior-posterior positions along the primitive streak), (2) final cell fate at 48 hours (representing distinct mesendodermal lineages), and (3) E- and N-cadherin expression (cadherin switching). They found that cells along the anterior-posterior axis of the primitive streak retain the potential to differentiate into various mesendoderm fates by 48 hours, but display distinct dynamics of E- and N-cadherin switching. The apparent mismatch between final cell fate and cadherin profiles led the authors to propose that cadherin switching and cell fate potential are not correlated during mesendoderm differentiation from hPSCs—unlike the stronger coupling seen during neuroectodermal differentiation. The authors conclude with E-cadherin knockout experiments, which show minimal impact on N-cadherin upregulation and cell fate, supporting the idea that cadherin switching and fate potential are decoupled in mesendodermal differentiation.

The authors leverage their strength in 2D human stem cell models and employ strategies to differentiate hPSCs into populations mimicking different regions of the primitive streak. They then assess cell fate potential and cadherin dynamics across these distinct states. The temporal dynamics of cell fate potential towards the mesendoderm cell fate and their dependence on cadherin switching are interesting questions. If properly addressed, the study could contribute significantly to the field. However, the current version of the manuscript falls short in several critical areas and requires major revisions.

Major Issues:

1. Throughout all figures, the authors draw conclusions based on gene expression comparisons (e.g., via violin plots) across various treatment conditions, but no statistical analysis is presented. Without appropriate statistics, it is difficult to assess whether the observed differences are significant and support the authors' claims.
2. The authors focus exclusively on E- and N-cadherin, which are central to EMT studies, but other cadherin subtypes are also known to be expressed during mesendoderm differentiation. Given that transcriptomic data are available for all cell types, the authors should report which cadherins are expressed in each condition. Are E- and N-cadherin the only highly expressed cadherins? If so, this should be clearly stated. Otherwise, the exclusion of other cadherins must be justified, or the authors should clarify that their focus is limited and that other cadherins could still play roles in fate determination.

3. The manuscript largely presents correlative data, which is a major limitation. The temporal mismatch between cadherin switching and cell fate commitment suggests a possible decoupling, but this remains speculative without genetic perturbation data. The E-cadherin knockout (KO) experiment is a useful step, showing that its absence has minimal effect on N-cadherin expression and cell fate. However, this is insufficient on its own. More informative would be E-cadherin overexpression or N-cadherin KO experiments, which could directly test whether loss of E-cadherin or gain of N-cadherin is required for differentiation. The E-cadherin KO alone merely accelerates an expected loss after Activin treatment and does not robustly establish functional independence between cadherin dynamics and fate decisions.

Minor Issues:

* The medium used for DE differentiation does not include FGF, yet Figure 1A shows high FGF and Wnt signaling in DE. This mismatch should be corrected.

* The authors state that N-cadherin and Snail upregulation vary among DE, PM, and LM. While Snail clearly differs, Figure S1 does not convincingly show variation in N-cadherin expression. All three lineages exhibit similar trends, with only LM showing slightly lower final N-cadherin levels.

* Figure 2Cii: Swap the axes so that CDX2 is on the horizontal axis and TBX6 is on the vertical, for easier comparison with Figure 2Ci.

* Figure 4: The claim of the figure title that cadherin switching is independent of fate commitment is too strong. The data show no simple correlation between cadherin profiles and cell fate, but a more complex relationship cannot be ruled out. To make such a strong statement, one would need to inhibit cell fate commitment entirely and demonstrate that cadherin switching still occurs independently—something the current study does not do. In the text, data in Figure 4 is mostly used to support the lack of correlation between E-cadherin down regulation and N-cadherin up-regulation, but not their relationship with cell fate commitment.

* Figure 5: The transcriptomic analysis shows that treatment conditions cluster by day 2 protocols, not day 1, which contradicts earlier claims in Figure 3. For example, PPS-DE(b) and MPS-DE(b) cluster with APS-DE(b) and DE(a)-DE(b), suggesting MPS can give rise to DE(b), contrary to the claim that MPS cells cannot express SOX17. Furthermore, Figure 5C shows SOX17 expression in MPS-DE(b), contradicting Figure 3B, where SOX17 is absent in the same condition. This inconsistency must be clarified.

First revision

Author response to reviewers' comments

We thank all the reviewers for their careful reading, positive comments, constructive criticisms, and helpful suggestions. We have performed extensive additional experiments and revised the manuscript in response to these suggestions and believe that the new manuscript is significantly improved. Below, reviewer comments can be found in blue italics with our response in plain text.

Comments from the Reviewers:

Reviewer 1:

The article by Zhu et al explores the dependence of cell potential and cadherin switching on primitive streak coordinates using in vitro differentiation of human pluripotent stem cells as a model. The main conclusions from this work are that cells with different primitive streak coordinates have a broad potential to form different primitive streak fates and that EMT/cadherin switching and fate acquisition can be uncoupled. The experiments have been carefully designed and quantified, and therefore the results are supported by the data. Moreover, the conclusions are novel. I support the publication of this article but have a few comments that hopefully will help improve the clarity of the manuscript:

1. The identity of the extra-embryonic mesoderm cells that are mentioned at the beginning of the result section is not clear and the manuscript has contradictory statements. In some places

the authors conclude that these cells are either amnion or extra-embryonic mesoderm, and in some other places they take for granted these are amnion cells. Moreover, they also mention these cells co-express E-Cadherin and N-Cadherin while amnion cells do not express N-Cadherin and extra-embryonic mesoderm cells do not express E-Cadherin. Finally, they state that both amnion cells and extra-embryonic mesoderm cells express GATA3 but this is probably not the case. Based on Pham et al, 2022 (in vitro) and Tyser et al, 2021 (in vivo) extra-embryonic mesoderm cells do not express GATA3. This is also the case in the human reference atlas built by the Lanner and Petropoulos groups (Zhao et al, 2024).

We agree with the reviewer that GATA3 is typically associated with amnion cell fates and that we were insufficiently clear in discussing these cells in the text. Our discussion reflects some ambiguity in the expression of GATA3 as recent data from human 2D gastruloids do show some GATA3 expression in the extraembryonic mesoderm which derives from the primitive streak (Chen Nat Methods 2025). In the same dataset, AP2A shows a larger expression difference between these populations, so we performed the same differentiation protocols and stained for AP2A. Our results show that cells co-express AP2A and GATA3 and we therefore conclude that these cells are amnion, which is consistent with the reviewer's remarks above. We have included this data in the manuscript in Figure S2 and revised the text to clearly refer to these cells as amnion cells. We agree that amnion cells do not typically express N-Cad, however, this is another example of the plasticity of these cell types to modify their cadherin expression, which is one of the main messages of our paper. We now explicitly make this point in lines 129-136.

2. The uncoupling of cell identity and Cadherin switching is very interesting. This is clearly demonstrated in Figure 5B. Nonetheless, the authors could perform a differential gene expression analysis to determine whether any other gene expression modules or signaling pathways are affected when cadherin expression is different within a given cellular identity.

We have now performed RNA sequencing on E-CAD KO cells and assayed the differences between KO and wildtype cells. These data confirm that (1) Each ECAD KO condition clusters with the corresponding wildtype condition, (2) a list of developmental genes do not have substantial expression changes upon loss of E-CAD, and (3) performing PCA on these samples shows each ECADKO sample aligns with its corresponding wildtype sample in PC1. We found that in the PCA there was a consistent shift in PC2 and identified a subset of genes that do change expression. These were, for the most part, not developmentally associated genes, and, for example, gene ontology in endoderm differentiation showed GO terms such as "mRNA splicing" and "glycolytic process" associated with the genes that change. Overall, this suggests that while loss of E-CAD may change the biology of these cells, it has minimal effects on differentiation. These data are now presented in Figure S20 and lists of differentially expressed genes and GO terms are given in Supplementary Tables 3 and 4.

3. Figure S1A shows E-Cadherin downregulation before over Snail1 expression. What may be triggering the initial E-Cadherin downregulation in this context? This could at least be discussed.

We agree that this is an interesting question. Our data with modified protocols show that E-Cad is rapidly downregulated when Activin signaling is highest on day 1, suggesting a more direct rapid regulation of E-Cad than through Snail. We now mention this in the discussion section in line 602-603.

4. The authors present some data that support the idea that E-Cadherin downregulation and N-Cadherin upregulation may be independent. However, this conclusion could be strengthened with additional experiments. For example, if Activin-A is necessary for E-Cadherin downregulation during the first 24 hours, what would happen if Activin-A is added on the second day? Would this promote N-Cadherin upregulation without any changes in E-Cadherin?

We performed experiments in which we did the APS-PM protocol but adding varying doses of Activin on day 2 rather than including the A83 inhibitor. In contrast to day 1 where Activin promoted E-CAD downregulation and N-CAD upregulation, on day 2, Activin had limited effects on N-CAD expression. Thus, at least in the APS-PM protocol, Activin primarily effects cadherin switching on the first day. This data is now included in Fig. S14. See the next point for additional

data that support our original claim.

Along similar lines, while the E-Cadherin KO is informative, the opposite experiment (E-Cadherin upregulation) would also be interesting. Would N-Cadherin expression be affected under this condition?

We performed new experiments in which we overexpressed a fusion of E-CAD and GFP in the standard differentiation protocols to DE, PM, and LM. We deliberately did not select for transgene expression to allow us to compare N-CAD and fate marker expression between GFP positive and negative cells. We did not see changes in N-CAD based on E-CAD overexpression in any of the protocols confirming the independence of N-CAD upregulation from E-CAD downregulation. These data can be found in Figure S22.

Relatedly, we have also now performed RNA sequencing of the E-CAD KO cells which further confirm that there are no differences in N-CAD expression upon loss of E-CAD. These data appear in Figure S20.

5. Both Wnt and Activin-A are necessary for E-Cadherin downregulation. The authors should at least discuss whether these two signals may be working independently or within the same pathway.

We thank the reviewer for this suggestion. A number of studies have established a requirement for WNT signaling in Activin induced differentiation as stimulation of the Activin/Nodal pathway alone functions in pluripotency maintenance. Thus, the combinatorial action of these two pathways is required rather than it being an additive effect. We have now included discussion of this point in lines 603-608.

6. E-Cadherin elimination typically triggers the upregulation of other cadherin types as a compensatory mechanism. For example, in the epidermis E-Cadherin depletion triggers P-Cadherin expression (Tinkle et al, 2004). As a control, the authors should check the expression of different cadherins in their E-Cadherin KO cells, as this could explain the intact apicobasal polarity that they report.

We performed RNA sequencing of the E-CAD KO cells in pluripotency and following differentiation to LM, PM, or DE. Examining cadherin expression in this data showed only limited differences between wildtype and knockout cells. The most notable differences were in PCDH8, which was upregulated in the E-CAD KO cells specifically in the APS-PM protocol, although the functional significance of this change is beyond the scope of the current manuscript. We include this data on cadherin expression in Figure S21. Results in line 480-488; Discussed in line 566-572.

Reviewer 2:

In this manuscript, the authors investigate the relationship between cell fates and the changes in E-Cadherin and N-Cadherin expression known to accompany differentiation in the primitive streak. The authors use a nice framework of hPSC differentiation to mimic cell fate allocation towards distinct antero-posterior identities and then use quantitative imaging to correlate cell fate markers and cadherin expression under various conditions.

By switching the signalling environment of the cells, they find that cell fate plasticity is maintained after 24h of induction except for cells induced towards mid and posterior streak identities which lose the ability to form the most anterior fate. They go on to show that E-Cadherin and N-cadherin are regulated independently from one another by signalling pathways and can be decoupled from cell fate acquisition. Finally, they show that Ecad knockout has no effect on PS cell fate.

Overall, the reasons for the dynamic regulation of Ecad and Ncad during gastrulation remain unclear. The manuscript by Ye Zhu and Aryeh Warmflash presents an interesting study that uses a nice approach to delineate the respective contributions of signalling and cell differentiation on EMT markers. I think the data will be of interest to the readership of Development but there are some areas that require improvement to meet the standards of the journal:

In my opinion, some claims are not fully supported by the data, in particular the notion that Ecad/Ncad regulation can be decoupled from cell fate. Data presentation would benefit from tightening up and the transcriptomic analyses should be further exploited to refine the message of the paper.

Major:

1) Cell fate characterisation:

** "Cells subjected to the LM protocol exhibited some heterogeneity and included a small subpopulation of amnion-like cells"*

The fact that this population displays heterogeneity is important for interpretation of the data shown later in the manuscript (see my comment number 2). The authors, tested the presence of amnion and cardiac progenitors (Fig S3) - which is useful - but pluripotent cells primed with WNT and FGF and then subjected to BMP may also produce other mesodermal derivatives such as PAX2+ intermediate mesodermal cells and TBX6+ PXM cells. It would be important to further characterise heterogeneity in this condition, using other stainings (TBX6, PAX2 for example). The authors could also exploit their RNAseq dataset shown later in the manuscript, focusing here on the fate markers expressed in each condition.

We thank the reviewer for raising this issue. We first clarified the identity of the amnion-like cells by staining for additional markers including AP2A and AP2C. This showed that cells that are positive for E- CAD and GATA3 are also positive for these markers which confirms their identity as amnion cells. These data can be found in Figure S2. Second, we note that nearly all cells in these cultures either express these amnion markers or express markers of lateral mesoderm such as ISL1 and HAND1 in the absence of the amnion markers (In Fig. 1Diii, only 0.3% of cells are double negative for ISL1 and HAND1) and thus we believe that these two fates make up the vast majority of the culture. Finally, as suggested, we also analyzed our RNAseq data for the markers mentioned by the reviewer. Both TBX6 and PAX2 are present at very low levels in the LM protocols and so we do not believe there are substantial amounts of cells adopting these fates. These data can be found in Figure S11.

** "In vivo, the amnion does not express N-Cad, and the extraembryonic mesoderm does not express E- Cad, but our results here indicate cells adopting one of these fates are capable of expressing both cadherins"*

This is interesting and it would be important to determine what exactly is this Ecad+/Ncad+ double positive population. Could this reflect asynchrony in the rate of differentiation in the culture? Is this population maintained at day 3? Please comment.

As noted above, we stained these conditions for additional markers and found that these cells co-express GATA3, AP2A, and AP2C which supports their identification as amnion cells (Figure S2,10). Thus, we believe that these cells are amnion-like cells with aberrant expression of N-CAD, consistent with the idea that the cadherin expression can vary within a given fate.

In general, we did not extend our differentiation protocols to the third day, as this would us to try various protocols on day 3, which is beyond the scope of our study. However, we do have data on subjecting the MPS-LM protocol to cardiac differentiation for an additional day, where we found that E-CAD/N-CAD double positive cells persist to this point. We reproduce this data here (Figure R1).

Fig R1. Cadherin expression profiles under cardiac differentiation. Scatter plots show co-

expression of E- and N-Cadherin at the single-cell level. Each dot represents one cell ($n =$ [the total number of cells per condition]); color indicates density of overlapping points. Dashed lines denote manually defined thresholds used for quadrant separation. Percentages indicate the proportion of cells in each quadrant. (2 independent experiments performed. 6 images per condition.)

2) Cell fate plasticity

In Fig3, the authors show that MPS and PPS lose potency to form DE. Is this really the case or are the cells delayed in redirecting to the DE fate? This could be addressed by looking at marker expression at day 3.

To answer this question, we extended the differentiation of the PPS-DE and MPS-DE conditions for an extra day in DE medium. The fraction of SOX17+FOXA2+ double positive DE cells remained low (<5%) so we conclude that MPS and PPS cells have indeed lost the potential to form DE. These data can now be found in Figure S6 and are mentioned in the corresponding text (line 198-202).

3) Correlation between fate and Ecad/Ncad expression:

One of the main messages of the authors is that fate and EMT markers can be decoupled. This idea rests mostly on the comparison between Fig 2 and Fig 4 "By day 2, the expression of all three of these markers can vary within cells of the same cell fate, as their expression, but not the cell fate, depends on the day 1 conditions." (Minor: it would be helpful if the authors could reference the corresponding figures here)

We now refer to the corresponding figures in text

I don't think this is well supported by the data: Fig 4A shows only marginal variation in Ncad levels in the fates redirected to LM. Furthermore Ncad seems to correlate closely with the levels of CDX2 in the fates redirected to PM (Fig 4A and 2B).

The main change occurs with Ecad levels which are elevated in PPS → LM, and TGFb- → LM. These conditions correspond to the situations where the population is heterogenous and enriched for ISL1+ cells (Fig. 2 iv). PPS→LM (higher ISL1 + higher Ecad), -TGFb →LM (higher ISL1 and higher Ecad).

These observations contradict the authors claim. Perhaps a more in depth analysis of their co-staining of Ecad with ISL1 (Fig S2) would help clarifying this point - Is there indeed a lack of correlation between these 2 markers at the single cell level? The same should be done with CDX2 and Ecad. Complementary to this, and to circumvent the technical limitations of image quantification of membrane markers, the authors could use their double reporter line, sort out the cells based on levels of Ecad/Ncad and check the levels of fate markers by qPCR.

We thank the reviewer for raising this important point. We agree with the reviewer that this is a complex issue as fate and cadherin switching both depend on signaling. Nonetheless we believe the data support that these processes are not directly coupled and so it is possible to find signaling combinations that lead to the same fate but very different cadherin levels. We have edited the text to reflect this more nuanced view and discussed this explicitly in the discussion section lines 574-585.

To directly address the reviewer's questions, we performed additional experiments performing costaining for E-CAD, N-CAD and a variety of fate markers in the PM and LM protocols with varying primitive streak induction on the first day. We found the following:

1. Cells induced to LM displayed a wide variety of E-CAD and N-CAD levels and these largely did not correlate with expression of markers of amnion or lateral mesoderm including GATA3, AP2A, HAND1, and ISL1. The main exception was AP2A which primarily showed high E-CAD expression levels, however, some AP2A+ cells also had lower E-CAD or high levels of N-CAD.
2. Cells differentiated to PM fates also showed widely varying E-CAD and N-CAD levels. Here, the majority of cells expressed CDX2 regardless of cadherin levels, while TBX6 expression was more restricted to a particular region of the E-CAD-N-CAD plane. Essentially all TBX6 cells were

N- CAD positive, however, some TBX6 cells had low levels of N-CAD, and TBX6+ cells ranged from not expressing E-CAD to expressing substantial levels of E-CAD.

Our data can be summarized as follows: all cell fates were consistent with substantial variations in cadherin levels. LM cells were consistent with any level of cadherin expression, TBX6+ PM cells all expressed some N-CAD but could express E-CAD, and all AP2A amnion cells expressed E-CAD but some expressed N-CAD. For the required expression (N-CAD in PM, E-CAD in amnion), levels were highly variable.

These data are now included as Figure S9-10 and discussed in the corresponding text line 264-284.

4) Influence of signalling on the levels of Ecad/Ncad:

*"During the initial 24 hours as cells exited pluripotency, Activin downregulated while BMP upregulated E- Cad"
I am not sure where is the evidence that BMP upregulates Ecad, the highest levels are found in the Ctrl where there is no BMP treatment.*

The reviewer is correct that pluripotent cells express high levels of E-CAD. Our statement here reflects the fact that the difference between APS and MPS is primarily the addition of BMP and MPS cells have higher levels of E-CAD than APS cells. Similarly, PPS differs from the -TGFb condition due to addition of BMP and has higher E-CAD levels. We have clarified this point in the corresponding text (line 235-236).

"both Snail and N-Cad expression were highest in the condition with both Activin and BMP treatment, suggesting that both signals contribute to upregulating these factors" I agree for Snail but this is not so clear for NCad.

We agree with the reviewer that while this is technically true, the difference is minor, and we have removed N-CAD from this statement (line 236-237).

"cells with APS yielded lower E-Cad and higher Snail and N-Cad compared to DE(a) (Fig. 4B-D)" I agree for Snail at day 1 but I can't see any significant difference at day 2.

We agree that these differences, while present, are smaller on day 2 and have clarified this point in the text (line 251-252).

Please comment on the three points above. Overall, it is difficult to decouple fate from signalling since signalling dictates fate. I think the authors should discuss this and clarify which data exactly demonstrate that they were able to uncouple the relative contribution of differentiation and signalling on EMT markers. Alternatively, they could moderate the claim "that EMT and cadherin switching can be decoupled from cell fate acquisition and are regulated independently by signaling pathways" and acknowledge the limitations of the study in the discussion section. It would also be useful if the authors could discuss how to reconcile the idea that Activin promotes Ecad downregulation while other authors have shown that endoderm (an activin-driven cell fate) maintains Ecad and only undergo partial EMT (Scheibner et al)

As we note above, we have edited the text to reflect the more nuanced view that although cadherin switching and cell fate are not directly coupled, some relationship between them exists due to their dependence on the same signaling pathways.

Regarding the Scheibner et al study, we agree that this is an interesting question. Our data are primarily consistent with a role of Activin signaling downregulating E-CAD on day 1 of differentiation while endoderm requires more prolonged Activin signaling and the relationship between Activin and cadherin switching shifts on later days (Please see our response to reviewer 1 question 4 where we describe new data related to this point and see the data in Figure S14). We now include a discussion of this point in the discussion section (Line 612-618).

5) Transcriptomic data "Transcriptome analysis confirms broad potential of PS progenitors" This

claim rests on the observation that hierarchical clustering regroups treatments on day 2 regardless of day 1 conditions. I am not sure if this is a strong argument to confirm the earlier results (Fig. 2-3). In particular, all PS populations further differentiated towards DE cluster together even though the authors argued that MPS and PPS lost ability to make DE. The transcriptomic profile of PPS-LM also seem to differ significantly from the other samples.

This dataset probably contains lots of useful information and a more in depth analysis is needed. For example, it would be helpful to see the DEG between different PS conditions further differentiated towards the same fate. Such analyses will clarify differences and similarities between treatments.

This is a valid point regarding clustering of transcriptomic data not being sufficient to establish potential of cells. We believe that the transcriptomic data together with our examination of key markers make a strong case for the broad potential of most PS subtypes. As we note in response to reviewer 3's specific question about this point below, the failure of PPS and MPS conditions to express key markers of endoderm such as FOXA2 and SOX17 likely indicates a loss of potential to adopt these fates. Fully establishing the potential of cells would require not only looking at markers but continuing to differentiate these cells to, for example, endoderm subtypes to establish their further potential. Although such studies would be very valuable, they are beyond the scope of our current manuscript. We now discuss the limitation of using markers to establish cell fate potential in the discussion section. (Lines 540-547)

6) Clarity, consistency and stats

** I found it difficult to understand the logical progression between figures at times. For example, in Fig 1, the authors introduce their differentiation scheme and characterise marker expression for DE, APS, and MPS (PPS is shown in the diagram but not tested in the actual experiment). Then, Fig 2A (about fate plasticity) drops DE, tests PPS and introduce a new TGFb-condition. It would improve consistency and completeness if all the differentiation conditions were introduced and characterised in Fig 1.*

We appreciate the reviewer's comments but prefer our original organization and we indicate the rationales for these choices here: Fig 1 aims to reproduce existing protocols and the PPS protocol was not included in the original Loh et al study. We did include PPS in the schematic here to show the spatial relationship between the PS positions. When we tested cell fate potential of mesoderm, which was done in Fig 2, it made sense to develop and include a PPS protocol. As endoderm is tested in a separate figure, it wasn't included here. Given that PPS, MPS, and APS primarily differ in the addition of the TGFb superfamily members Activin and BMP4, the -TGFb protocol is included as a control in Fig 2, but it is not intended to mimic any particular part of the PS, and therefore is not included in the schematic in Fig 1.

** The manuscript includes quantifications and most data points are shown - which is great - but the legends do not state how many independent experiments were performed and there are no statistical test in any of the figures. This is needed to assess the robustness of the data and help with interpretation. The scatter plots could also include proportions as % in the quadrants of the plots. If these represent triple staining on different plots, it would help to show the proportions of each subpopulation in a table or in a bar chart or pie chart.*

We thank the reviewer for these suggestions and have now included tests for statistical significance in all figures, indicate percentages of cells in the quadrants of figures, and included statements of number of independent experiments in the legends.

7) It is interesting that the Ecad KO does not alter cell fate. The manuscript would benefit from additional discussion regarding potential compensation mechanisms upon Ecad KO.

We have performed RNA sequencing of the E-CAD KO line and looked at expression of other cadherins to check for potential compensation. While the expression of most cadherins is unchanged, PCDH8 is upregulated specifically in the APS-PM protocol. We have included this data in Figure S21 and included discussion of this in lines 480-488 .

Minor:

** Fig. 1 - the difference between the DE and APS conditions is a bit unclear, is it simple that FGF is not included in the DE condition or are the concentrations in WNT and Activin different?*

We thank the reviewer for pointing this out. The DE(a) protocol differs from APS not only in the absence of FGF but also in the levels of Activin (higher in DE) and WNT (lower in DE). We have revised Fig. 1Aii by updating the table to show “++” for high activation, “+” for activation, “-:” for inhibited, and “/” for no factor added. We also added a legend below the table for clarity.

Below are the DE(a) and APS protocols

	Activin A (ng/mL)	Chir (μ M)	FGF (ng/mL)	medium
DE(a)	100	2	/	Essential 6
APS	30	4	20	Essential 6

** Fig. S1 and S2 are mentioned after S3 in the text.*

We have now reordered the supplementary figures so they appear in order in the text.

** I found it inconvenient to switch between figures to compare fate markers and EMT markers (Fig 2 and 4 for example). The authors might consider reorganising the manuscript to show the results of the switch to PM on one figure containing both fate and EMT markers and then another figure showing the switch to LPM.*

We understand and appreciate the reviewer’s point of view, however, upon reflection, we still feel that our manuscript is best organized thematically rather than according to differentiation protocol. Organizing following this suggestion would still require flipping back and forth to compare, for example, markers in PM and LM treatments.

** Fig 4B has AntiGFP signal, this is not commented and not explained what GFP represents here*

We used the anti-GFP antibody to enhance the signal of the cerulean-CAAX membrane marker. This is now explained in the text (line 230-232). To avoid confusion, we have updated the figure label from “Anti- GFP” to “CAAX” and clarified this in the legend.

Reviewer 3:

Using in vitro differentiation of human pluripotent stem cells (hPSCs) toward mesendoderm fate, the authors investigate the interdependence of (1) intermediate cell state at 24 hours (corresponding to different anterior-posterior positions along the primitive streak), (2) final cell fate at 48 hours (representing distinct mesendodermal lineages), and (3) E- and N-cadherin expression (cadherin switching). They found that cells along the anterior-posterior axis of the primitive streak retain the potential to differentiate into various mesendoderm fates by 48 hours, but display distinct dynamics of E- and N-cadherin switching. The apparent mismatch between final cell fate and cadherin profiles led the authors to propose that cadherin switching and cell fate potential are not correlated during mesendoderm differentiation from hPSCs—unlike the stronger coupling seen during neuroectodermal differentiation. The authors conclude with E-cadherin knockout experiments, which show minimal impact on N-cadherin upregulation and cell fate, supporting the idea that cadherin switching and fate potential are decoupled in mesendodermal differentiation.

The authors leverage their strength in 2D human stem cell models and employ strategies to differentiate hPSCs into populations mimicking different regions of the primitive streak. They then assess cell fate potential and cadherin dynamics across these distinct states. The temporal dynamics of cell fate potential towards the mesendoderm cell fate and their dependence on cadherin switching are interesting questions. If properly addressed, the study could contribute significantly to the field. However, the current version of the manuscript falls short in several critical areas and requires major revisions.

Major Issues:

1. Throughout all figures, the authors draw conclusions based on gene expression comparisons (e.g., via violin plots) across various treatment conditions, but no statistical analysis is presented. Without appropriate statistics, it is difficult to assess whether the observed differences are significant and support the authors' claims.

We thank the reviewer for raising this issue and have now added tests of statistical significance to all figures.

2. The authors focus exclusively on E- and N-cadherin, which are central to EMT studies, but other cadherin subtypes are also known to be expressed during mesendoderm differentiation. Given that transcriptomic data are available for all cell types, the authors should report which cadherins are expressed in each condition. Are E- and N-cadherin the only highly expressed cadherins? If so, this should be clearly stated. Otherwise, the exclusion of other cadherins must be justified, or the authors should clarify that their focus is limited and that other cadherins could still play roles in fate determination.

This is a valid point and we now include analysis of the expression of other cadherins from transcriptomic data in Figure S20. We also performed additional transcriptomic measurements on the E-CAD knockout cell line to test for compensation of other cadherins and include this data also in Figure S20. Finally, we mention in the discussion section (line 566-572) that we focus our attention on the classical cadherin switch between E- and N-CADHERIN but that other cadherins may play roles in differentiation.

3. The manuscript largely presents correlative data, which is a major limitation. The temporal mismatch between cadherin switching and cell fate commitment suggests a possible decoupling, but this remains speculative without genetic perturbation data. The E-cadherin knockout (KO) experiment is a useful step, showing that its absence has minimal effect on N-cadherin expression and cell fate. However, this is insufficient on its own. More informative would be E-cadherin overexpression or N-cadherin KO experiments, which could directly test whether loss of E-cadherin or gain of N-cadherin is required for differentiation. The E-cadherin KO alone merely accelerates an expected loss after Activin treatment and does not robustly establish functional independence between cadherin dynamics and fate decisions.

We thank the reviewer for raising this issue and agree that this would enhance the manuscript. We performed experiments with E-CAD overexpression and find essentially no effect on cell fate markers and minimal effect on N-CAD expression. We also characterized the ECAD-KO cell line in more detail by performing RNA sequencing of cells. These data are now included in Figure S20-21.

Minor Issues:

* The medium used for DE differentiation does not include FGF, yet Figure 1A shows high FGF and Wnt signaling in DE. This mismatch should be corrected.

We thank the reviewer for this point and have now corrected this in Fig. 1A:

1. In Fig. 1Ai, we placed FGF in parentheses to indicate that not all fate induction protocols require FGF (e.g., DE(a) does not include FGF).
2. We have revised Fig. 1Aii by updating the table to show ++ for high activation, + for activation, - for inhibited, and / for no factor added. We also added a legend below the table for clarity.

* The authors state that N-cadherin and Snail upregulation vary among DE, PM, and LM. While Snail clearly differs, Figure S1 does not convincingly show variation in N-cadherin expression. All three lineages exhibit similar trends, with only LM showing slightly lower final N-cadherin levels.

We have updated the text in this section to be more precise about the claims about N-CAD expression.

* Figure 2Cii: Swap the axes so that CDX2 is on the horizontal axis and TBX6 is on the vertical, for easier comparison with Figure 2Ci.

We have made this change.

** Figure 4: The claim of the figure title that cadherin switching is independent of fate commitment is too strong. The data show no simple correlation between cadherin profiles and cell fate, but a more complex relationship cannot be ruled out. To make such a strong statement, one would need to inhibit cell fate commitment entirely and demonstrate that cadherin switching still occurs independently—something the current study does not do. In the text, data in Figure 4 is mostly used to support the lack of correlation between E-cadherin down regulation and N-cadherin up-regulation, but not their relationship with cell fate commitment.*

We have modified the title of Figure 4 to soften this claim (line 288). See also our responses to reviewer 2 question 3 above, where we note that we have changed the text to reflect a more nuanced view of the relationship between fate and cadherin switching. That is, while they are not directly coupled and cadherin expression can vary significantly within cells of the same fate, the dependence of both on the same signaling pathways means that they are not fully independent either. We describe in detail the limitations that this places on cadherin expression within each fate in Figure S9-10 and the associated text (line 264-284).

** Figure 5: The transcriptomic analysis shows that treatment conditions cluster by day 2 protocols, not day 1, which contradicts earlier claims in Figure 3. For example, PPS-DE(b) and MPS-DE(b) cluster with APS- DE(b) and DE(a)-DE(b), suggesting MPS can give rise to DE(b), contrary to the claim that MPS cells cannot express SOX17. Furthermore, Figure 5C shows SOX17 expression in MPS-DE(b), contradicting Figure 3B, where SOX17 is absent in the same condition. This inconsistency must be clarified.*

This is a fair point and reflects the complexity in one measurement reflecting the overall transcriptome, which may bear some similarity between e.g. MPS-DE(b) and DE(a)-DE(b), and the other reflecting individual crucial fate markers, which are not expressed in MPS-DE(b). Nonetheless, we think it is reasonable to state that cells cannot fully be adopting endoderm fates if they are not expressing key genes such as SOX17 and FOXA2. See our response to reviewer 2 question 5. We now discuss and clarify this discrepancy in the text regarding the transcriptomic data (Line 321-329). We also discuss the limitations of determining cell fate potential based solely on expression in the discussion (lines 540-547).

Second decision letter

MS ID#: dev.204807R1

MS TITLE: Dependence of cell fate potential and cadherin switching on primitive streak coordinate during differentiation of human pluripotent stem cells

AUTHORS: Ye Zhu and Aryeh Warmflash

Dear Dr Warmflash,

I have now received all the referees reports on the above manuscript, and have reached a decision. The referees' comments are appended below.

The overall evaluation is very positive and we would like to publish a revised manuscript in Development, provided that the remaining referees' comment can be satisfactorily addressed. Specifically, Reviewer #3 has asked for additional statistical evaluation. Please add this information to Fig. S3 and other relevant figure panels in the manuscript, and adjust the associated main text as necessary. If you do not agree with this suggestion, then please explain clearly why this is so in your response.

Reviewer 1*Advance summary and potential significance to field*

The authors have taken on board the suggestions of the reviewers. They have done a great job addressing all the concerns that were raised. I am very satisfied with the responses provided and I think the manuscript is a nice contribution to the Development field.

Reviewer 2*Advance summary and potential significance to field*

I am satisfied with the revisions. I think the text is clearer, the data has been reinforced and the more nuanced discussion has improved the manuscript. This is an interesting human stem cell study that brings novel information about the co-dependence (or absence thereof) of EMT and cell fate during gastrulation . I recommend publication in Development. Congrats to the authors.

Reviewer 3*Advance summary and potential significance to field*

This is a revised manuscript of a previous submission. The authors address the dependence between E- and N-cadherin switching and the cell states in Day 1 & 2 during differentiation of human pluripotent stem cells towards mesendoderm fates. They found that the cells' E-cadherin status is largely independent from their fate differentiation process.

Comments for the author

I appreciate the authors' efforts to address my concerns, including adding more statistical analysis, discussing the role of additional cadherins and cadherin compensation, modifying unsupported conclusions, and performing E-cadherin over-expression to test the independence of cell fates. Despite the authors' effort to include statistics, I still find it insufficient. For example, the authors claim that "The timing and level of N-CAD upregulation and EMT marker expression varied among the three lineages within the first day of induction." To me, Fig. S3A show N-CAD goes up slightly in all conditions at day 1. A statistical test comparing the N-CAD level at 1 day would address this, but the test is missing. The authors did add statistical test for data within the same plot. But throughout the manuscript, there are conclusions made by comparing data across plots, and these statements should be supported by statistics as well (especially if they are not visually obvious).

Second revisionAuthor response to reviewers' comments*Comments from the Reviewers:**Reviewer 1:*

The authors have taken on board the suggestions of the reviewers. They have done a great job addressing all the concerns that were raised. I am very satisfied with the responses provided and I think the manuscript is a nice contribution to the Development field.

Reviewer 2:

I am satisfied with the revisions. I think the text is clearer, the data has been reinforced and the more nuanced discussion has improved the manuscript. This is an interesting human stem cell study that brings novel information about the co-dependence (or absence thereof) of EMT and cell fate during gastrulation. I recommend publication in Development. Congrats to the authors.

We thank reviewers 1 and 2 for their efforts to improve our manuscript and for their positive comments on our revision.

Reviewer 3:

This is a revised manuscript of a previous submission. The authors address the dependence between E- and N-cadherin switching and the cell states in Day 1 & 2 during differentiation of human pluripotent stem cells towards mesendoderm fates. They found that the cells' E-cadherin status is largely independent from their fate differentiation process.

I appreciate the authors' efforts to address my concerns, including adding more statistical analysis, discussing the role of additional cadherins and cadherin compensation, modifying unsupported conclusions, and performing E-cadherin over-expression to test the independence of cell fates.

We thank the reviewer for these positive comments.

Despite the authors' effort to include statistics, I still find it insufficient. For example, the authors claim that "The timing and level of N-CAD upregulation and EMT marker expression varied among the three lineages within the first day of induction." To me, Fig. S3A show N-CAD goes up slightly in all conditions at day 1. A statistical test comparing the N-CAD level at 1 day would address this, but the test is missing. The authors did add statistical test for data within the same plot. But throughout the manuscript, there are conclusions made by comparing data across plots, and these statements should be supported by statistics as well (especially if they are not visually obvious).

We have gone through the manuscript carefully to look for missing statistical tests. We found that statistics were missing from Figures 1 and S3 (the figure the reviewer highlighted), and have added them. We thank the review for calling this to our attention.

Regarding comparison between figures, we have searched for cases where claims were made based on multiple figures and could not find any that were lacking statistics. There are several cases where we refer to multiple figures, but each reference is self-contained with controls and statistics. For example: "Cells treated with DE(a) activated BRA, the PS/mesodermal progenitor marker, albeit at the lowest level compared to the other PS subtypes (Fig. 3B). At this stage, expression of the endodermal marker SOX17 remained undetectable. Upon extending the treatment of different PS subtypes with DE(b) in the second day, the DE(a)-DE(b) and APS-DE(b) groups showed similar co-expression of the endodermal markers SOX17 and FOXA2 (Fig. S5ii)." (lines 161-165) These statements are derived from two figures, however, each makes a separate point, one regarding the expression of DE markers on the first day, and the other regarding DE markers on the second day. In each case controls and statistics are provided within the figure. We do not believe any central claims in our paper are currently lacking statistics.

Third decision letter

MS ID#: dev.204807R2

MS TITLE: Dependence of cell fate potential and cadherin switching on primitive streak coordinate during differentiation of human pluripotent stem cells

AUTHORS: Ye Zhu and Aryeh Warmflash

Dear Dr Warmflash,

Thank you for submitting your revised manuscript to Development. I am satisfied with the changes that you have made in response to Reviewer 3's remaining comment. We have identified a small number of omissions that require correction please. Once you have submitted a suitably revised version, I would be happy to accept the manuscript. The changes needed are:

Fig. S1E: This panel is not described in the figure legend or mentioned in the main text of the article.

Fig. S3A and B: Please add information to the figure legend about the statistical test(s) used and description of the symbols shown (e.g. "**") to denote significance. This would bring the legend in line with the other figures.

Fig. S11 mentions in the legend that error bars are included in these panels. I cannot see any error bars. It could be that they are small and below the resolution of the figure, but I wanted to highlight this in case the error bars have been omitted by mistake.

I look forward to receiving a revised manuscript from you in due course.

Third revision

Author response to reviewers' comments

Fig. S1E: This panel is not described in the figure legend or mentioned in the main text of the article.

In general, we have included signaling data on modulating the key pathways for each protocol, for example, WNT for paraxial mesoderm and BMP for lateral mesoderm. This is additional data on modulating WNT in the lateral mesoderm protocol. It does not fit easily within the flow of the manuscript, and we have therefore opted to remove it from the current manuscript.

*Fig. S3A and B: Please add information to the figure legend about the statistical test(s) used and description of the symbols shown (e.g. "**") to denote significance. This would bring the legend in line with the other figures.*

Thank you for bringing this to our attention. This information has been added.

Fig. S11 mentions in the legend that error bars are included in these panels. I cannot see any error bars. It could be that they are small and below the resolution of the figure, but I wanted to highlight this in case the error bars have been omitted by mistake.

Thank you for bringing this to our attention as this was indeed an error. We now include the corrected version with error bars.

Fourth decision letter

MS ID#: dev.204807R3

MS TITLE: Dependence of cell fate potential and cadherin switching on primitive streak coordinate during differentiation of human pluripotent stem cells

AUTHORS: Ye Zhu and Aryeh Warmflash

Dear Dr Warmflash,

I am happy to tell you that your manuscript has been accepted for publication in Development, pending our standard publication integrity checks.